# Nascent peptide-induced translation discontinuation in eukaryotes impacts biased amino acid usage in proteomes

Yosuke Ito[1,4], Yuhei Chadani [2,4] ✉, Tatsuya Niwa [1,2], Ayako Yamakawa[1], Kodai Machida[3], Hiroaki Imataka[3] & Hideki Taguchi [1,2] ✉

Robust translation elongation of any given amino acid sequence is required to shape proteomes. Nevertheless, nascent peptides occasionally destabilize ribosomes, since consecutive negatively charged residues in bacterial nascent chains can stochastically induce discontinuation of translation, in a phenomenon termed intrinsic ribosome destabilization (IRD). Here, using budding yeast and a human factor-based reconstituted translation system, we show that IRD also occurs in eukaryotic translation. Nascent chains enriched in aspartic acid (D) or glutamic acid (E) in their N-terminal regions alter canonical ribosome dynamics, stochastically aborting translation. Although eukaryotic ribosomes are more robust to ensure uninterrupted translation, we find many endogenous D/E-rich peptidyl-tRNAs in the N-terminal regions in cells lacking a peptidyl-tRNA hydrolase, indicating that the translation of the N-terminal D/E-rich sequences poses an inherent risk of failure. Indeed, a bioinformatics analysis reveals that the N-terminal regions of ORFs lack D/E enrichment, implying that the translation defect partly restricts the overall amino acid usage in proteomes.

Life depends on the functions of proteins. Myriads of proteins in living organisms are synthesized by ribosomes, and robust translation elongation of any given amino acid sequence is required to shape proteomes. The translation mechanisms of ribosomes, including the essential properties of the peptidyl transferase center (PTC) and the exit tunnel, are well conserved in all living organisms. The ribosome complex processively synthesizes the nascent polypeptide chain without detachment, despite the large conformational changes during the elongation cycle. The elongation by the translating complex progresses, decoding any codon on the open reading frame (ORF) of an mRNA until the ribosome arrives at a stop codon. Essentially, the ribosomes decode any codon on ORFs. The lengthening nascent polypeptides pass through the ribosomal exit tunnel[1], which encloses 30–40 amino acids of the growing nascent polypeptidyl-tRNA[2,3].

Recent studies have revealed that the translation elongation rate is non-constant[3]. The dynamics of translating ribosomes are affected not only by the scarcity of tRNAs in corresponding codons and mRNA secondary structures[4–7] but also by specific interactions between the ribosome exit tunnel and the nascent chain[3,8,9]. The tunnel-nascent chain interactions result in altered translational speeds in both bacteria and eukaryotes[10–13]. Mechanistically, the electrostatic interaction between negatively charged ribosomal RNAs and positively charged nascent polypeptides could cause ribosomal pausing[14–16]. For instance, the translation of proline stretches reportedly causes elongation pausing, which is alleviated by eIF5A[17]. Recent global studies on eIF5A using the ribosome profiling method showed that eIF5A also affects the elongation of tripeptide motifs enriched with negatively charged amino acids (Asp: D and Glu: E), in addition to proline stretches[18,19]. The translation of aberrant mRNAs causes elongation pausing and

[1]School of Life Science and Technology, Tokyo Institute of Technology, Yokohama 226-8503, Japan. [2]Cell Biology Center, Institute of Innovative Research, Tokyo Institute of Technology, Yokohama 226-8503, Japan. [3]Graduate School of Engineering, University of Hyogo, Himeji, Hyogo 671-2280, Japan. [4]These authors contributed equally: Yosuke Ito, Yuhei Chadani. ✉e-mail: ychadani@bio.titech.ac.jp; taguchi@bio.titech.ac.jp

ribosome collision, and is subsequently eliminated by ribosome-associated quality control (RQC) systems in eukaryotes[20,21].

In addition to translation pausing, nascent chains even destabilize the translating ribosome from within. We recently found that *Escherichia coli* ribosomes are destabilized by a class of amino acid sequences in nascent chains, typically containing D/E-runs such as ten consecutive glutamates (10E), stochastically leading to premature termination termed intrinsic ribosome destabilization (IRD)[22,23]. The peptidyl-tRNAs produced by IRD are cleaved by peptidyl-tRNA hydrolase (Pth). Pth, an essential enzyme in bacteria, recycles the peptidyl-tRNAs outside the ribosome complex by cleaving the ester bonds between the peptides and tRNAs. Because the translation of negatively charged sequences is risky, there are several counteracting mechanisms to prevent premature termination. A high concentration of magnesium, which can stabilize the ribosome complex, counteracts IRD. Interestingly, this $Mg^{2+}$-associated phenomenon of IRD provides a physiological impact. In addition, most potential IRD sequences in the middle of ORFs remain shielded by the tunnel-spanning nascent polypeptide preceding the IRD sequences[23].

Since translation mechanisms are conserved in all domains of life, it is possible that IRD takes place even in eukaryotic cells. Compared to bacterial ORFs, eukaryotic ORFs contain many consecutive D/E sequences, which are considered to extend the protein function in eukaryotic organelles such as the nucleus. For example, essential genes encoding more than 20 D/E-runs are present in budding yeast. Here, we show that the premature translation termination induced by nascent chains enriched in D or E residues around N-terminal regions also occurs in eukaryotes, as revealed by an in vivo reporter assay and a reconstituted cell-free translation system. The accumulation of the resultant abortive peptidyl-tRNAs is toxic, since yeast cells lacking Pth cannot grow when IRD-prone sequences are overexpressed. We identify many endogenous IRD-derived peptides by a mass spectrometry (MS) analysis of the RNA fractions in the Pth-deletion strain. A bioinformatics analysis unveiled that proteomes have a biased amino acid distribution to avoid the premature termination risk induced by IRD. Thus, the translation elongation process disfavors a subset of amino acid sequences harboring D/E-rich sequences in their N-terminal regions.

## Results

### Premature termination during the translation of consecutive negatively charged amino acids in the N-terminal regions

We recently found a noncanonical translation dynamics termed IRD, where nascent chains enriched in D/E residues destabilize the translating *E. coli* ribosome and stochastically terminate translation in a context-dependent manner[22,23]. To determine whether such premature translation termination also takes place in eukaryotic cells, we constructed a reporter system based on a dual-luciferase assay to monitor the potential IRD in *Saccharomyces cerevisiae* (Fig. 1a). Firefly luciferase (Fluc) and *Renilla* luciferase (Rluc, as an internal expression control) were expressed under a galactose-inducible bidirectional promoter. Fluc was fused to GFP, a homo-decamer of amino acids (10X; 10D, 10E, 10N, 10Q, and 10T), a linker peptide, and a self-cleaving peptide, T2A (Fig. 1a). If the translation of the consecutive D/Es such as 10D induced translation abortion, then the ratio of the activity of Fluc to that of Rluc, defined as the translation continuation (TC) index, would decrease.

The TC indices for the 10D and the 10E fused with full-length GFP ($GFP_{FL}$) were 0.9–1.1, when normalized by those of 10N and 10Q, respectively, to eliminate the effect of the negative charges in the motifs (Fig. 1b), showing uninterrupted translation continuation in these 10D and 10E constructs. Since IRD in *E. coli* tends to occur very early in the coding sequence[22], we then shortened the GFP moiety to 5 ($GFP_5$) and 30 ($GFP_{30}$) amino acids from the N-terminus. Strikingly, the TC indices decreased depending on the GFP length in both 10D/10N and 10E/10Q: a shorter GFP length led to a lower TC index (Fig. 1b). The TC indices of 10D/10T, in which the 10T sequence is encoded in $(ACG)_{10}$, frameshifted

codons of $(GAC)_{10}$ for the 10D, also depended on the GFP length as in the case of 10D/10N, confirming that the amino acid sequence, but not the RNA sequence, is critical for the reduction (Supplementary Fig. 1a).

The reduced luciferase activities in the 10D and 10E constructs might be due to mRNA quality control systems, including no-go decay[24,25] (NGD) and nonsense-mediated decay[26,27] (NMD). To address this, we used the *hel2Δ* and *nmd2Δ* strains, with impaired NGD-mediated ribosome dissociation and NMD-mediated clearance of aberrant mRNA, respectively. The 10D-dependent reduced TC indices in the constructs harboring shorter GFP moieties were not affected in both strains (Fig. 1c), confirming that NGD and NMD are not responsible for the reduced translation. In addition, the TC indices of $GFP_5$–10D (or $GFP_{30}$–10D) were not altered even in the strain bearing the temperature-sensitive *hyp2*, which encodes the eIF5A protein[28] (Fig. 1c), after we verified the inefficient translation of 10 consecutive prolines in the *hyp2*[ts] strain at a semi-lethal temperature, 33 °C, using the dual-luciferase system (Supplementary Fig. 1b, c). Taken together, the major translation quality control systems known to date are not involved in the consecutive D/E-dependent translation attenuation.

### D/E-run-dependent peptidyl-tRNA accumulation in a reconstituted cell-free translation system

In *E. coli*, a reconstituted cell-free translation system (*E. coli* PURE system) was used to demonstrate that the polyacidic residue-dependent premature termination occurs even with the minimum factors required for translation. Therefore, we adopted a human factor-based reconstituted cell-free translation system, the HsPURE system[29], to ultimately determine whether the decreased translation is intrinsically induced by the nascent peptide chain. We constructed fusion genes encoding GFP ($GFP_5$, $GFP_{30}$, or $GFP_{FL}$)−10X-Fluc (N-terminal 84 a.a. fragment). CrPV-IRES[30], which allows translation initiation without initiation factors, was used to minimize the length of peptide translated prior to the D/E-runs. These fusion mRNAs were translated with the HsPURE system, and the translated products were separated by neutral pH SDS-PAGE, which can analyze intact peptidyl-tRNA species (Fig. 1d)[8,22,23]. All of the $GFP_{FL}$-containing constructs produced full-length polypeptides, regardless of whether 10X was 10D or 10 N (Fig. 1e, lanes 13, 16). In the constructs harboring $GFP_{30}$, the 10D construct produced several bands (lane 7), whereas the major band from the 10 N construct was the full-length polypeptide, which is insensitive to RNase (lane 10). The largest band in $GFP_{30}$–10D (lane 7) was sensitive to RNase (lane 9), indicating that the band is a peptidyl-tRNA. In addition, we found that one of the *S. cerevisiae* Pths (Pth2p: the reason why Pth2p was chosen is shown below) cleaved the resultant peptidyl-tRNA (lane 8), indicating that the peptidyl-tRNAs are accessible to Pth2p. In the construct harboring the shorter GFP sequence ($GFP_5$), the translation produced more RNase- and Pth-sensitive peptidyl-tRNA species (lane 1). The TC indices calculated by quantifying the band intensities in the HsPURE system (Fig. 1f) were consistent with those obtained from the luciferase reporter assay in yeast (Fig. 1b).

These in vivo and in vitro results demonstrated that the translation cessation by the N-terminal D/E-runs in eukaryotes does not require any factor other than the essential translation components, eventually resulting in the accumulation of premature translation products.

### Behavior of eukaryotic ribosomes translating the D/E-runs

We next evaluated whether the 80 S ribosome translating the D/E-runs shows the IRD-like features observed in the *E. coli* ribosome. First, we tested the $Mg^{2+}$ concentration dependence of the translation abortion, since IRD in *E. coli* is suppressed by high $Mg^{2+}$ concentrations, which stabilize the ribosome subunit association[23,31,32]. Additional $Mg^{2+}$ in the cell-free translation system partially alleviated the premature translation termination in $GFP_5$–10D (Fig. 2a) and $GFP_{30}$–10D (Supplementary Fig. 2a), but did not affect the translational pausing in hCMV uORF2 (Supplementary Fig. 2b), suggesting

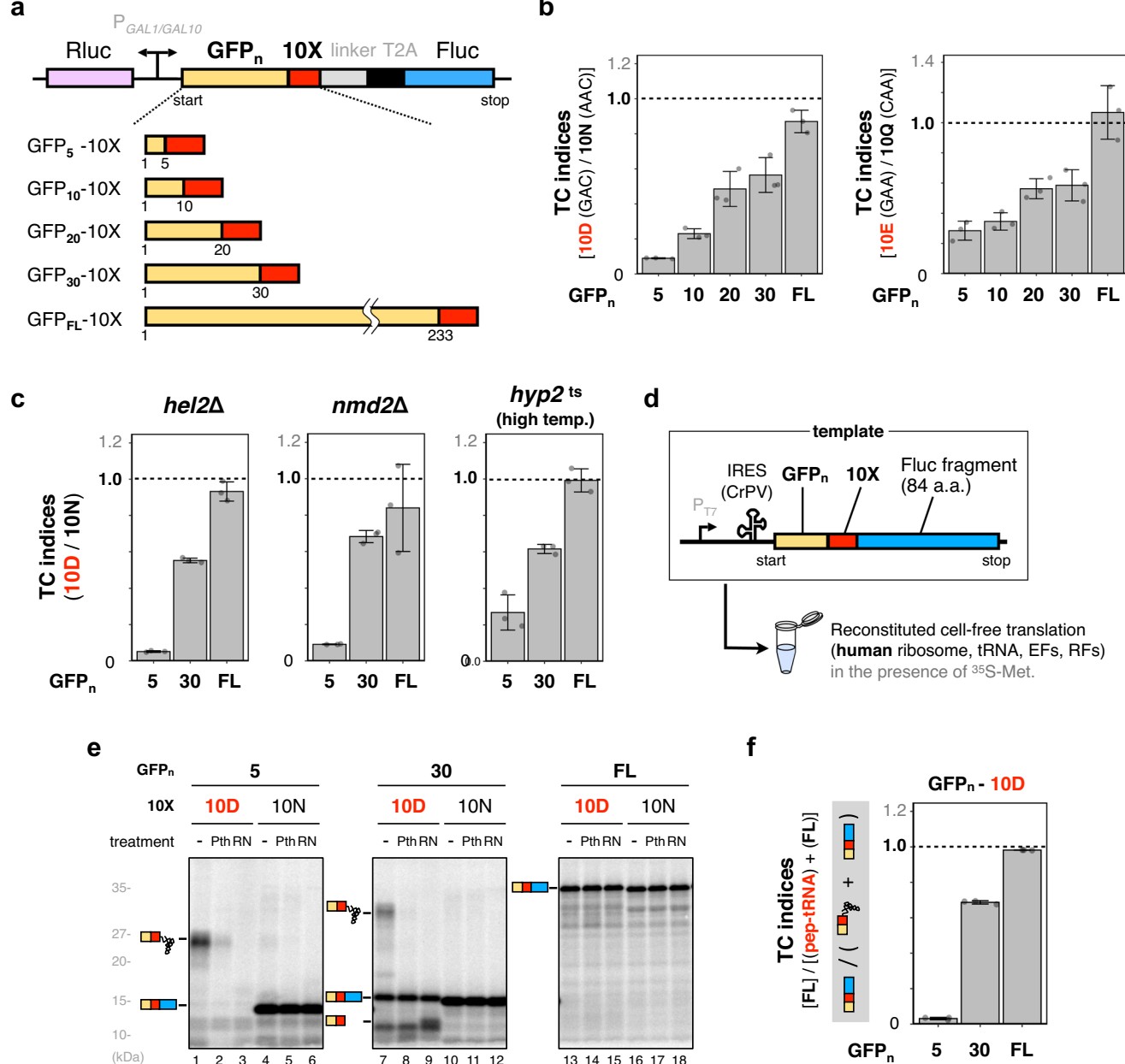

**Fig. 1 | Consecutive negatively charged amino acids in the N-terminal regions prematurely terminate translation. a** Schematic representation of the dual-luciferase assay. Homo-decamers of amino acids (10X) were inserted between varying lengths of the N-terminal region of GFP (GFP$_n$; $n = 5$, 10, 20, 30, and 233 as full-length: FL) and a linker. Firefly (Fluc) and *Renilla* (Rluc) luciferases were used as reporter enzymes to investigate the translation continuation and an internal translation control, respectively. The *GAL1-GAL10* promoter is a galactose-inducible bidirectional promoter. T2A is the self-cleaving peptide from *Thosea asigna* virus 2A[77]. **b** The translation efficiencies of ORFs harboring 10X in *S. cerevisiae* were evaluated by the reporter system shown in **a**. After induction by galactose, the translation efficiencies were measured as the relative activity of Fluc to that of Rluc. Translation continuation (TC) indices were calculated by the following formula: {Fluc activity/Rluc activity (10D)} / {Fluc activity/Rluc activity (10 N)}. Error bars indicate standard deviations (SD) from three biological replicates. The codons encoding 10X are indicated in parentheses. **c** TC indices evaluated using the reporter system in the *hel2Δ*, *nmd2Δ*, and *hyp2* temperature-

sensitive (*ts*) strains are shown. The *hyp2*$^{ts}$ cells were grown at 33 °C instead of 30 °C, the normal cultivation temperature. Error bars indicate standard deviations (SD) from three biological replicates. **d** Schematic representation of the fusion gene, GFP$_n$-Methionine-10X-truncated Fluc, for the cell-free assay. A methionine codon was inserted between GFP and 10X to label the translation product with $^{35}$S-methionine. **e** The fusion genes were translated using the human factor-based reconstituted cell-free translation system (HsPURE system) including $^{35}$S-methionine, followed by puromycin treatment. The products were further treated with a peptidyl-tRNA hydrolase (yeast Pth2p: *Pth*) as indicated and separated by neutral pH SDS-PAGE with optional RNase A (*RN*) treatment. Radioactive bands were developed by using an imaging plate. Peptidyl-tRNAs and tRNA-cleaved polypeptides are indicated by schematic labels. **f** TC indices calculated from the band intensities in **e**. The TC index was defined by the following formula: {full-length polypeptide (FL: 10D)} / {full-length polypeptide (FL: 10D) + peptidyl-tRNA (10D)}. Error bars indicate standard deviations (SD) from three technical replicates.

that the translation discontinuation is somehow associated with the destabilization of the ribosome, similar to that in *E. coli*.

To directly evaluate whether the 80 S ribosome is destabilized during the translation of the D/E-runs, we fractionated translating ribosomes by sucrose density gradient ultracentrifugation (SDG). The GFP$_5$−10D peptidyl-tRNA was detected in the fraction containing the 80S complex (Fig. 2b), indicating that, in contrast to the prokaryotic ribosomes[23], eukaryotic ribosomes are not split by the translation of a

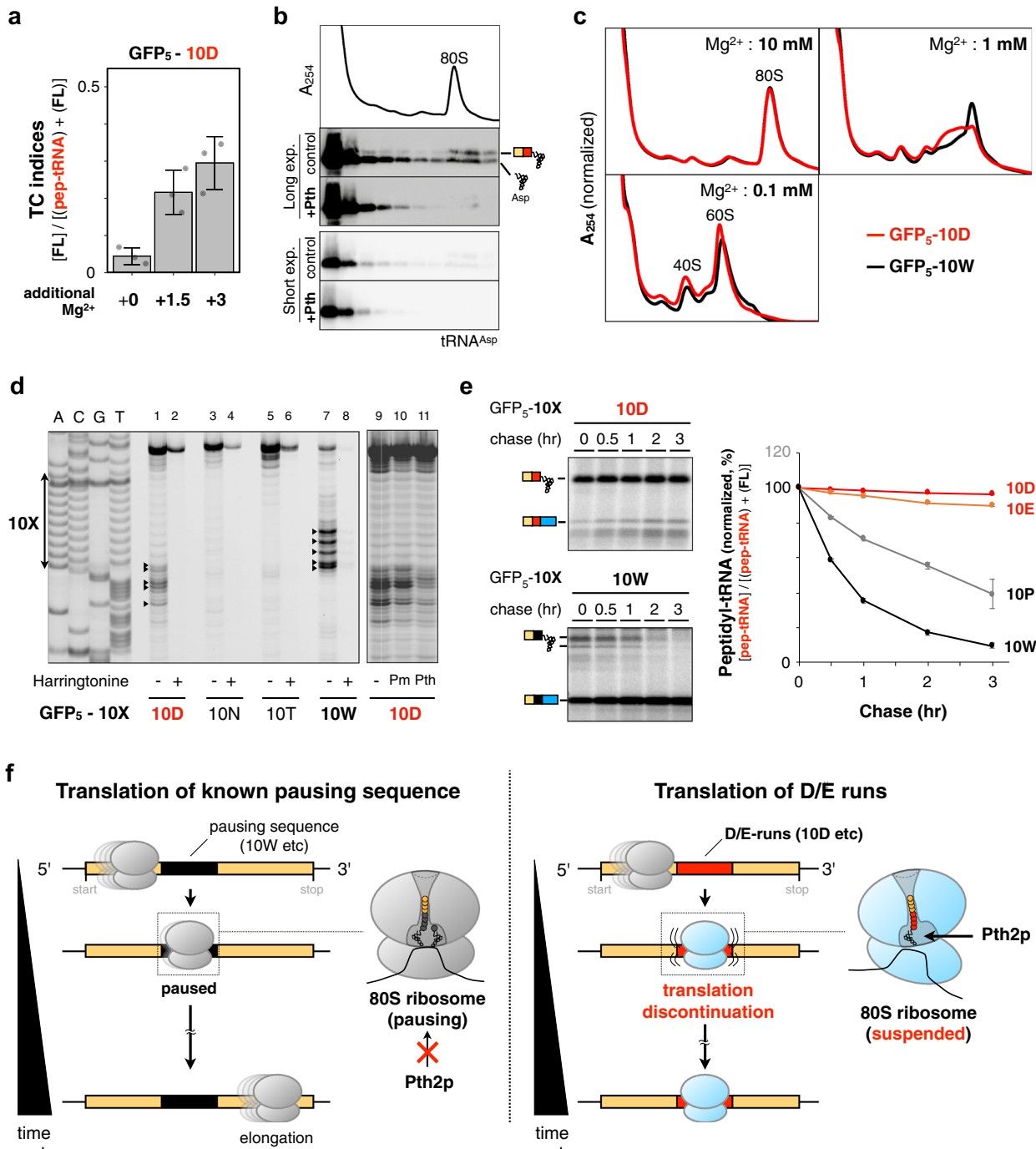

polyacidic sequence. In addition, Pth2p treatment diminished the co-migration of the peptidyl-tRNA and the 80S ribosome (Fig. 2b, +Pth). It should be noted that the peptidyl-tRNAs within the intact 80S complex were not cleaved by Pth2p when the ribosomes accommodated pausing sequences such as consecutive Lys or Trp residues (Supplementary Fig. 2c). Further support for the destabilization was provided by the Mg$^{2+}$ concentration dependence in the SDG analysis. Decreasing the Mg$^{2+}$ concentration made the ribosomes translating the GFP$_5$–10D more unstable compared to those translating the GFP$_5$–10W (Fig. 2c). These results suggest that GFP$_5$–10D peptidyl-tRNA intrinsically destabilizes and alters the 80S complex conformation, but unlike *E. coli*, does not necessarily split the 80S ribosome complex.

To assess this further, we performed toeprint analyses. As shown in Fig. 2d, we observed several reverse transcripts, indicating the presence of the ribosomes on the 10D mRNA sequence, as on the 10 W pausing sequence (Fig. 2d, lanes 1, 2 7, and 8, summarized in Supplementary Fig. 2d)[33]. Interestingly, these reverse transcripts disappeared by the Pth2p treatment (Fig. 2d, lanes 9–11). Taken together with the SDG analysis, the ribosomes translating 10D-runs are likely to maintain the 80 S complex in an altered state and could be resolved by Pth2p under the in vitro conditions.

This assumption was also supported by the comparison of the "half-life" of GFP$_5$-10X peptidyl-tRNAs. Most of the $^{35}$S-methionine-labeled GFP$_5$–10D or GFP$_5$–10E peptidyl-tRNAs remained even after a 3 h chase with unlabeled methionine, while the GFP$_5$–10P or GFP$_5$–10W pausing sequences gradually proceeded to the downstream translation (Fig. 2e). These results together indicated that the translation of the polyacidic sequence destabilizes the 80S complex

**Fig. 2 | Characterization of the 80S ribosome translating the DE-runs. a** Effect of $Mg^{2+}$ on the 10D sequence-dependent translation attenuation. The $GFP_5$−10D sequences were translated by the HsPURE system supplemented with additional $Mg^{2+}$ as indicated, and individual TC indices were calculated. Error bars indicate standard deviations (SD) from three technical replicates. **b** The HsPURE system mixture translating the $GFP_5$−10D peptidyl-tRNA was fractionated by sucrose density gradient ultracentrifugation, separated by SDS-PAGE, and detected by northern blotting. Distributions of the ribosomes were monitored by $A_{254}$ measurement. The chemiluminescence images with long- or short-time exposure were shown. Translation mixture was treated with yeast Pth2p (Pth) if indicated. A representative of three independent experiments is shown. **c** The HsPURE system mixture translating the $GFP_5$−10D or $GFP_5$−10W sequence was fractionated as shown in **b** except that the $Mg^{2+}$ concentration in the fractionation buffer was arranged (10, 1, or 0.1 mM). A representative of two independent experiments is shown. **d** Toeprint analysis. The HsPURE system mixture was directed by the $GFP_5$−10D, 10 N, 10 T, or 10 W template (lanes 1–8), in the presence or absence of 10 mM harringtonine, respectively. Reaction mixtures were then subjected to reverse transcription using a downstream, fluorescent primer. Dideoxy sequencing reactions were also primed by the same primer (lanes A, C, G, and T). The reaction mixture of $GFP_5$−10D was treated with puromycin (lane 10) or Pth2p (lane 11) before the reverse transcription. A representative of three independent experiments is shown. **e** Inactivation of the 80 S ribosome by the translation of D/E-runs. The $GFP_5$−10D, 10E, 10 P, or 10 W mRNA was translated by the HsPURE system including $^{35}S$-methionine for 1 h. Then excessive unlabeled methionine (2 mg/mL) was added to stop the incorporation of $^{35}S$-methionine and further incubated for the indicated time. The $^{35}S$-labeled translational products at each time point were withdrawn and monitored by gel electrophoresis (left). The amount of peptidyl-tRNA was calculated as following formula {peptidyl-tRNA (10X)} / {full-length polypeptide (10X) + peptidyl-tRNA (10X)}, and their normalized values (0 min: 100) were plotted with standard deviations (SD) from three technical replicates. **f** Schematic representation for a plausible model based on biochemical experiments. Left, translation of known pausing sequences such as 10 W pauses but resumes the translation elongation. Pth2p cannot cleave the corresponding peptidyl-tRNA in the 80 S ribosome. Right, translation of polyacidic sequences alters the conformation of the 80 S ribosome in an altered state, and then causes translation discontinuation. Pth2p can cleave the peptidyl-tRNA associated with the 80 S ribosome.

and somehow alters its conformation, eventually aborting the elongation (Fig. 2f).

### Endogenous ORFs harboring D/E-rich sequences in the N-terminal regions induce premature termination

We next examined yeast ORFs harboring multiple D/E residues in their N-terminal regions, to determine whether such D/E-rich sequences in endogenous ORFs induce premature termination. We found 169 ORFs in *S. cerevisiae* that contain 4 or more D/E residues in the first 10 residues after the initial methionine. One ORF (*JIP3*) with 8 D/Es was the maximum (Fig. 3a). In addition to this ORF with 8 D/Es, we selected two and three ORFs with 6 and 4 D/Es, respectively, in the N-terminal regions to investigate whether premature termination occurs during the translation of these candidates (Fig. 3a). The dual-luciferase reporter assay revealed that more than 60% of the translation of the ORF with 8 D/Es was discontinued prematurely (Fig. 3b). We also observed a relatively high rate of discontinuation during the translation of the two ORFs with 6 D/Es. In contrast, only one ORF with 4 D/Es (*RTF1*) exhibited high discontinuation efficiency (Fig. 3b).

We also performed the HsPURE in vitro translation to evaluate the accumulation of abortive peptidyl-tRNAs for these six endogenous ORFs (Fig. 3c). The cell-free translation of *JIP3* harboring 8 D/Es in the N-terminal region predominantly produced RNase- and Pth-sensitive peptidyl-tRNA species (Fig. 3d). We also observed the accumulation of such peptidyl-tRNAs during the translations of two ORFs with 6 D/Es and one ORF with 4 D/Es (*RTF1*), albeit to lesser extents as compared to the ORF with 8 D/Es (Fig. 3d). The translation reactions of other ORFs with 4 D/Es (*ANB1*, *NMD2*) hardly produced any band that was sensitive to RNase or Pth (Fig. 3d), consistent with the corresponding data obtained by the dual-luciferase reporter assay (Fig. 3b). Collectively, these results revealed that the endogenous ORFs harboring D/E-rich sequences in the N-terminal regions also induce the translation discontinuation, which is roughly proportional to the number of D/Es in the regions, as shown in the previous study[23]. In addition, the higher $Mg^{2+}$ concentration abrogated the accumulation of peptidyl-tRNA in the 8 D/E-containing JIP3p (Supplementary Fig. 3a), further supporting the idea that IRD is involved in the accumulation of the peptidyl-tRNA, leading to the translation abortion during the translation of endogenous ORFs.

The above results demonstrated that the N-terminal D/E-rich sequences abort translation. In contrast, when the D/E-runs were translated after $GFP_{FL}$, most of the IRD was circumvented even in the translation of $GFP_{FL}$−20D (or −20E) (Supplementary Fig. 3b). Since the *E. coli* ribosome was unable to translate $GFP_{FL}$−20D (or −20E) due to IRD (Supplementary Fig. 3c), the eukaryotic ribosome is more robust and continues translation elongation.

### D/E-runs-derived peptidyl-tRNAs are cleaved by Pth2p in vivo

The peptidyl-tRNA species produced by translation discontinuation should be resolved to maintain the processibility of the translation system. In *E. coli*, the peptidyl-tRNAs generated by premature translation termination are cleaved by Pth (peptidyl-tRNA hydrolase) to avoid the scarcity of tRNAs[34,35]. So far, *S. cerevisiae* has two known Pths, encoded by *PTH1* and *PTH2*[36,37]. In addition, recent studies revealed that Vms1p also cleaves the peptidyl-tRNA to release the peptide attached to the CCA 3' end of tRNA[38–41]. If one of them is involved in the cleavage of the peptidyl-tRNAs, then the accumulation of undigested peptidyl-tRNAs in a strain lacking the responsible gene would be potentially toxic. Accordingly, we examined whether the overexpression of the D/E-runs affected the growth of the deletion strains. After we confirmed that the three single deletion strains, *pth1Δ*, *pth2Δ*, and *vms1Δ*, were as viable as the wild-type strain, fusion genes encoding GFP-10X, in which GFP sequences ($GFP_5$, $GFP_{30}$, or $GFP_{FL}$) are followed by 10X (10D or 10 N), were expressed under a galactose-inducible promoter (Fig. 4a). Expression of the genes in all combinations did not affect the viabilities of the *pth1Δ* and *vms1Δ* strains, and the wild-type (Fig. 4a). In contrast, the expression of genes with truncated GFPs followed by 10D ($GFP_5$−10D and $GFP_{30}$−10D), but not those followed by the 10 N or 10 T series, in the *pth2Δ* strain was lethal (Fig. 4a, Supplementary Fig. 4a). The expression of the $GFP_5$− or $GFP_{30}$−10E instead of that of the 10D counterparts also exhibited toxicity (Supplementary Fig. 4b), indicating that the D/E-runs are critical for the harmful phenotype. The lethal phenotype of *pth2Δ* upon the expression of $GFP_5$−10D was rescued by the expression of Pth2p or *E. coli* Pth, but not Pth1p (Fig. 4b).

The results of this phenotype assay strongly support the assumption that the accumulation of the IRD-derived peptidyl-tRNAs in the *pth2Δ* strain is toxic. To obtain direct evidence for the accumulation of peptidyl-tRNAs in the *pth2Δ* strains, we conducted a northern blot analysis to detect the peptidyl-tRNA species. Total RNA fractions were individually isolated from wild-type and *pth2Δ* strains by phenol extraction, and then electrophoresed, electroblotted, and probed with an anti-tRNA$^{Asp}$ oligonucleotide. We detected a higher molecular weight band, in addition to the tRNA$^{Asp}$ band, in the total RNA isolated from the *pth2Δ* strain with the $GFP_5$−10D overexpression (Fig. 4c, lane 6). This higher molecular weight band was not observed in the northern blot for the $GFP_5$−10N using the tRNA$^{Asp}$ oligonucleotide, confirming that the consecutive Asp residues are responsible for the higher molecular weight band. This band in the *pth2Δ* lysate was sensitive to treatments with Pth proteins, Pth2p or *E. coli* Pth, or heat under high pH conditions, further supporting that it is the abortive product of the $GFP_5$−10D translation (Fig. 4d).

Final proof for the accumulation of the peptidyl-tRNAs in the *pth2Δ* strain was provided by the MS analysis. The MS analysis of the

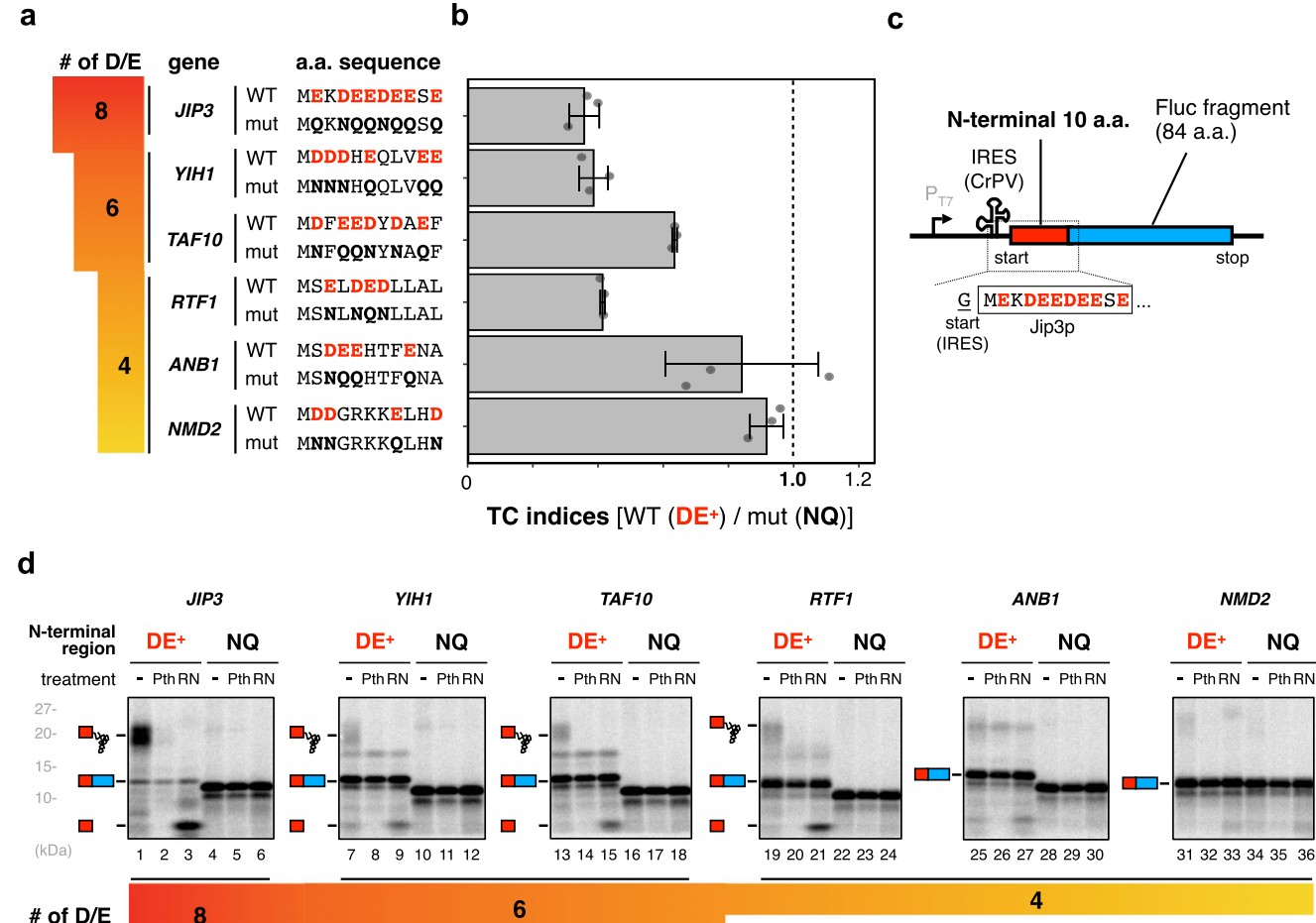

**Fig. 3 | Translation of endogenous genes that have multiple Asp/Glu (D/E) residues in the N-terminal regions. a** The list of representative genes with 8, 6, and 4 D/Es in the N-terminal 11 amino acids. The N-terminal sequences (*wt*) are shown with the mutant sequences (*mut*), in which D/Es are substituted by N/Qs. **b** The N-terminal 10 amino acid sequences except for the first methionine were inserted instead of the 10X in the dual-luciferase reporter construct shown in Fig. 1a. TC indices were calculated by the following formula: {Fluc activity (*wt*) / Rluc activity (*wt*)} / {Fluc activity (*mut*)/Rluc activity (*mut*)}. Error bars indicate standard deviations (SD) from three biological replicates. **c, d** The N-terminal 11 amino acids of the genes with multiple D/Es were fused with truncated Fluc (**c**). **d** The genes were translated using the HsPURE system, as described in Fig. 1e.

phenol-extracted RNA fraction of the *pth2Δ* strain expressing GFP₅−10D identified several unique peptides derived from the peptidyl-tRNAs: MS signals for GFP₅−6D, 7D, 8D, 9D, and 10D in the GFP₅−10D sequence (Fig. 4e, Supplementary Fig. 4c). Such stochastic production of various numbers of negatively charged residues in the peptidyl-tRNAs also occurs in IRD in *E. coli*[23]. We also expressed GFP₅−10N in the *pth2Δ* strain, but scarcely detected MS signals derived from the peptidyl-tRNA (Supplementary Fig. 4d).

Taken together, we concluded that the nascent negatively charged amino acid stretches at the N-terminal regions induce IRD, stochastically terminate translation, and eventually produce peptidyl-tRNAs in the eukaryotic translation system. In the wild-type yeast strain, the peptidyl-tRNAs are cleaved by Pth2p to prevent the harmful accumulation of abortive products. Accordingly, the Pth2p protein would be essential for cell viability under stress conditions, where the abortive peptidyl-tRNAs are highly accumulated.

**Accumulation of endogenous peptidyl-tRNAs with D/E-enriched peptides in *pth2Δ* cells**

A series of experiments in the *pth2Δ* strain prompted us to broadly explore the endogenous ORFs that prematurely terminate the translation in an IRD-dependent manner. To this end, we developed a new MS method to analyze endogenously accumulated abortive peptidyl-tRNAs in the *pth2Δ* lysate. The method is a combination of the isolation of the RNA fraction containing peptidyl-tRNAs[42], the heat- and high pH-induced cleavages of the peptide moieties from the peptidyl-tRNAs[43], and the identification of the resultant peptides by MS-based shotgun proteomics with tryptic digestion (Fig. 5a).

We identified more than 300 peptides from the RNA fraction in the *pth2Δ* lysate prepared from mid-log cultured cells (Fig. 5b, Supplementary Fig. 5a, Supplementary Data 1). Among them, ~100 peptides overlapped with the peptide set in the wild-type lysate, whereas ~200 peptides were specific to those in the *pth2Δ* cells (Fig. 5b, Supplementary Fig. 5a, Supplementary Data 1). Importantly, most of the *pth2Δ*-specific peptides disappeared upon RNase treatment during the preparation, indicating that they were derived from the peptidyl-tRNAs (Fig. 5c, Supplementary Data 1). We also identified more than 300 peptides from the RNA fraction in the *pth2Δ* lysate prepared from late-log cultured cells and ~200 *pth2Δ*-specific peptides (Supplementary Fig. 5b, Supplementary Data 1), suggesting that the *pth2Δ*-specific peptides are produced independently of the growth phases.

The *pth2Δ*-specific peptides derived from the peptidyl-tRNAs have notable distinguishing features. First, the *pth2Δ*-specific peptides were strongly enriched at the N-terminal regions of the ORFs, as compared to the wild-type-specific peptides: about 80% of the identified peptides were mapped within the 40 amino acids from the start methionine in the annotated ORFs (Fig. 5d, Supplementary Fig. 5c). Since the ~40 amino acid residues are considered to be accommodated in the ribosome exit

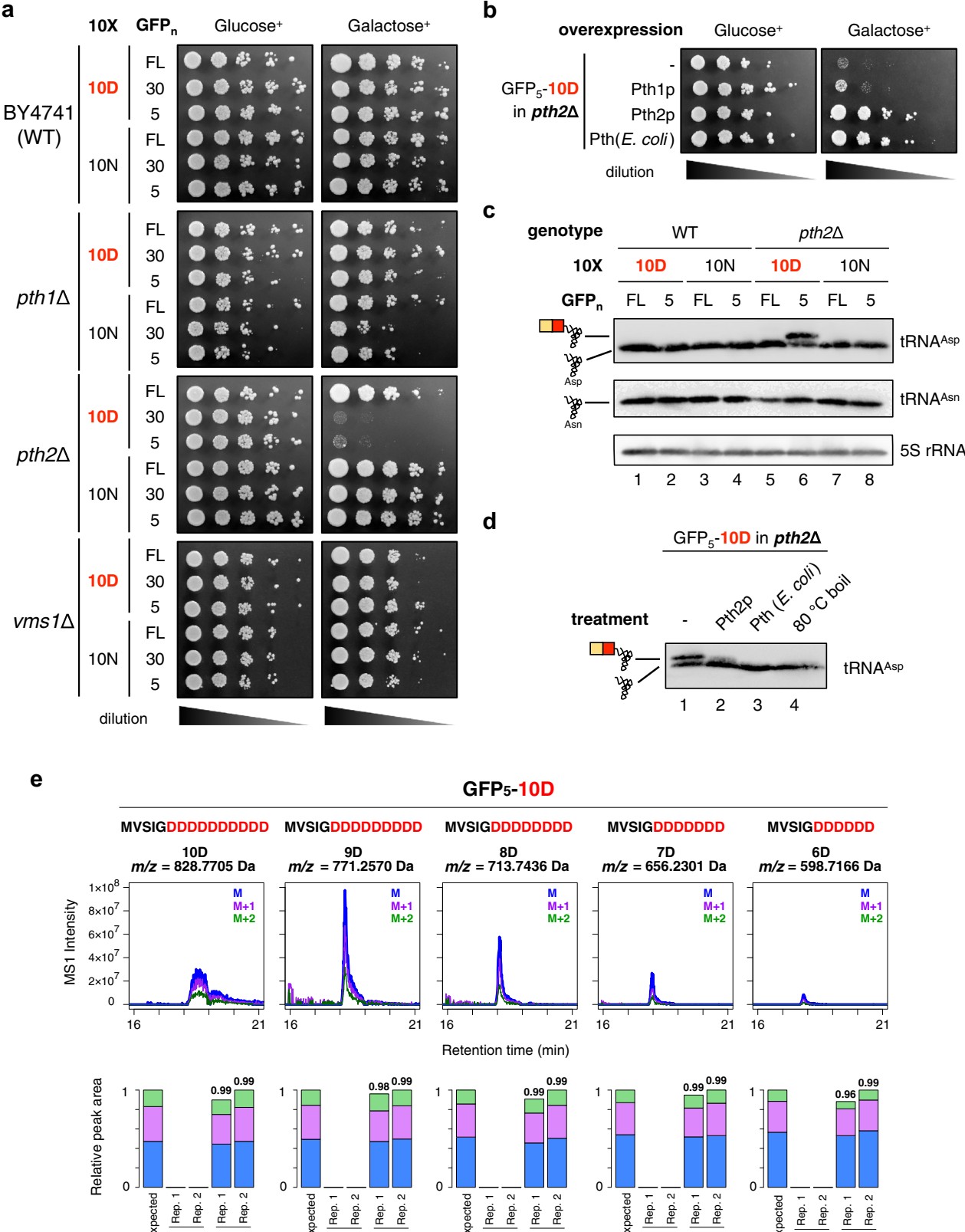

tunnel, the enrichment suggests that the *pth2Δ*-specific peptides are produced by premature termination before the N-terminal portions fully occupy the tunnel. Second, substantial fractions (56/220 and 23/167 for mid-log samples) of the *pth2Δ*-specific peptides had no Lys (K) or Arg (R) residues at their C-termini, as determined by a semi-tryptic peptide search (C-terminal non-K/R peptides, Fig. 5e, Supplementary Fig. 5d).

Since the peptides were digested by trypsin before the LC-MS measurement, these C-terminal non-K/R peptides were considered to be the C-termini of the peptide fragments derived from the peptidyl-tRNAs. Third, these *pth2Δ*-specific C-terminal non-K/R peptides tended to have a D/E-enriched C-terminus (Fig. 5f, g, Supplementary Fig. 5e, f). These features together indicate that the premature termination events are

**Fig. 4 | Pth2p, one of the peptidyl-tRNA hydrolases, is responsible for hydro-lyzing the peptidyl-tRNAs derived from the D/E-rich sequence-induced pre-mature termination. a** Wild-type (*WT*), *pth1Δ*, *pth2Δ*, and *vms1Δ* strains harboring the plasmids for the expression of the 10X-containing ORFs shown in Fig. 1a were spotted on SD (Glucose⁺) or SG (Galactose⁺) plates lacking uracil and cultivated at 30 °C for 2 days. Cultures were 10-fold serially diluted before spotting. **b** Complementation assay of Pth variants. The *pth2Δ* mutant harboring the GFP₅-10D construct showing the lethal phenotype, the transformed empty vector and plasmids carrying *pth1*, *pth2*, and *pth* from *E. coli*, were spotted on SD or SG plates lacking uracil and leucine and incubated at 30 °C for 2 days. **c** Northern blot analysis with anti-tRNA^Asp and tRNA^Asn probes showing tRNA^Asp with or without a short peptide (peptidyl-tRNA or tRNA, respectively) depending on strains and constructs. The results of 5S rRNA are also shown as a loading control. The result represents one of the two similarly conducted experiments. **d** Northern blot ana-lysis with the anti-tRNA^Asp probe, showing tRNA^Asp with or without a short peptide

depending on the hydrolysis treatment. RNA extract was incubated with 1 μM of purified Pth2p (Pth2p, at 30 °C for 20 min), 1 μM of purified *E. coli* Pth {Pth (*E. coli*), at 37 °C for 20 min}, or was heated under alkaline conditions (80 °C boil, at 80 °C for 20 min)[43]. The result represents one of the two similarly conducted experi-ments. **e** (Upper) Extracted ion chromatograms of the peptide fragment expressed from the GFP₅-10D sequence in the phenol-extracted RNA fraction of the *pth2Δ* strain. Each panel represents extracted ion chromatograms of MS1 intensity derived from a GFP₅-(X)D peptide. A chromatogram of monoisotopic ion is depicted in blue, and its +1 and +2 isotopes are depicted in purple and green, respectively. The specific *m/z* value of the monoisotopic ion is shown at the top. (Lower) Relative peak areas of MS1 chromatogram in the phenol-extracted RNA fraction of the wild-type (WT) and *pth2Δ* strain (two technical replicates). Areas of monoisotopic ions, +1 isotopic ions, and +2 isotopic ions are depicted in blue, purple, and green, respectively.

induced by IRD at the N-terminal region. A representative ORF meeting these features is SPT*8*. We identified six peptides from the N-terminus of Spt8p (1-MDEVDDILINNQVVDDE-17, 1-MDEVDDILINNQVVDDEE-18, 1-MDEVDDILINNQVVDDEED-19, 1-MDEVDDILINNQVVDDEEDD-20, 1-MDEVDDILINNQVVDDEEDDE-21, and 1-MDEVDDILINNQVVDDEEDDEE-22), reflecting the stochastic nature of the premature termination within the D/E-rich peptide (Fig. 5g, Supplementary Fig. 5f). In addition, the *pth2Δ*-specific C-terminal K/R peptides tended to have D/E-rich sequences within their N-terminal 100 amino acids (Supplementary Fig. 5g–i), implying that parts of these C-terminal K/R peptides are also associated with IRD.

To verify that the premature termination actually occurs during the translation of the genes with peptidyl-tRNAs detected in the *pth2Δ* strain, we examined candidate ORFs in vitro and in vivo (Fig. 6). Translations of three ORFs (*PIB2*, SPT*8*, and *SENS4*) using the HsPURE system resulted in the D/E-dependent accumulation of Pth-sensitive peptidyl-tRNA species (Fig. 6b), confirming that the premature termi-nation of these ORFs indeed occurs in the reconstituted cell-free translation. In addition, the overexpression of these three ORFs in the *pth2Δ* strain induced lethal phenotypes in a D/E residue-dependent manner (Fig. 6c), probably reflecting the detrimental effect of the abortive peptidyl-tRNAs on the cell viability. These results further confirmed that the translation of the N-terminal D/E-rich sequences in the endogenous mRNAs can cause premature translation termination.

**Proteomes avoid negatively charged amino acid clusters in the N-terminal regions**

The above results implied that the translation of D/E-rich amino acids in the N-terminal regions poses an inherent risk of premature trans-lation termination, from bacteria to human. Accordingly, we assumed that during evolution, organisms would have avoided the translation of D/E-rich amino acids occurring very early in the coding sequences. Therefore, we conducted a bioinformatics analysis to investigate the distribution of the D/E-rich sequences in proteomes.

We extracted 100 amino acids from the N-terminal, middle, and C-terminal regions of all known yeast ORFs (5540 ORFs), and then counted the numbers of proteins with three or more D/E residues in each ten-residue-moving window (Fig. 7a). Strikingly, we found that the number of proteins with ≥3 D/E residues in the N-terminal regions was low from the N-terminus to 30–40 amino acids, which is similar to the length of the ribosome tunnel. The N-terminal regions are critical in this trend, since there was no such reduction in the middle or C-terminal regions (Fig. 7a). The trend for the disfavored D/E usage was (i) specific to the D/E residues since there is no such low frequency for either N/Q or K/R residues (Fig. 7b and Supplementary Fig. 6a), (ii) more obvious in the analysis with ≥5 D/E residues (Supplementary Fig. 6b), and (iii) not due to a bias in nucleic acid sequences, since there is no such low frequency in the frameshifted DNA sequences (Sup-plementary Fig. 6c).

Further analyses using protein abundance data[44] revealed that the bias for the disfavored D/E residues at the N-terminal regions was mainly attributed to abundant proteins (Fig. 7c). This tendency is reasonable, since the translation of abundant proteins harboring N-terminal D/E-rich sequences would be more problematic or wasteful.

In addition to the yeast proteome, we analyzed the D/E usages in those of other organisms (Supplementary Fig. 7). The overall trend, including the human proteome, was to disfavor the D/E-rich sequences in the N-terminal regions, although the degrees of the bias varied. However, there were some proteomes with no apparent bias, such as *Arabidopsis thaliana* and the archaeon *Pyrococcus furiosus*. Collec-tively, the N-terminal D/E-rich peptide-dependent premature termi-nation, which is an inherent translation defect, could affect the amino acid distribution in a variety of proteomes.

## Discussion

We previously discovered a novel noncanonical ribosome behavior, IRD, in *E. coli*[23], but it was not obvious that IRD could be applied to a eukaryotic translation system. Based on the following overlapping features revealed in this study, we conclude that IRD also occurs in eukaryotes. First, nascent polypeptide chains with consecutive D/E residues in the N-terminal regions eventually led to the premature termination of translation elongation. Second, the translation of such D/E-rich amino acid sequences makes peptidyl-tRNAs a Pth2p-accessible state without splitting of the 80 S complex, indicating that the integrity of the translating ribosome is impaired. Third, such altered 80 S ribosome complex is inactive for processing the elonga-tion and is prone to dissociate under Mg²⁺-depleted conditions. Fourth, the stabilization of the translating ribosomes by increasing the Mg²⁺ concentration[23] or by the tunnel-occupying nascent polypeptide[22] alleviates the translation discontinuation, reflecting that the stability of the ribosome is associated with the premature termi-nation. Fifth, the translation discontinuation occurred in a human factors-based reconstituted cell-free translation system, indicating that only the essential translation factors are sufficient for the trans-lation discontinuation. Finally, the premature termination was sto-chastic during the translation of sequences enriched in negatively charged residues, as exemplified by the various aborted products, including 6 to 10D in the GFP₅–10D sequence identified by MS. Since the translation elongation mechanism is generally well conserved among all kingdoms of life, the conservation of this noncanonical ribosome dynamics is not surprising. In particular, the ribosome tun-nel and the PTC region are well conserved[2]. We thus conclude that IRD is a conserved property of the ribosomes from *E. coli* to eukaryotes.

Nevertheless, there are notable differences in IRD between *E. coli* and eukaryotes. First, the eukaryotic ribosomes are more robust to continuing the translation elongation of the D/E-runs. In *E. coli*, the stronger IRD motifs located in the internal regions of the ORF induce the translation abortion in a context-dependent manner. In

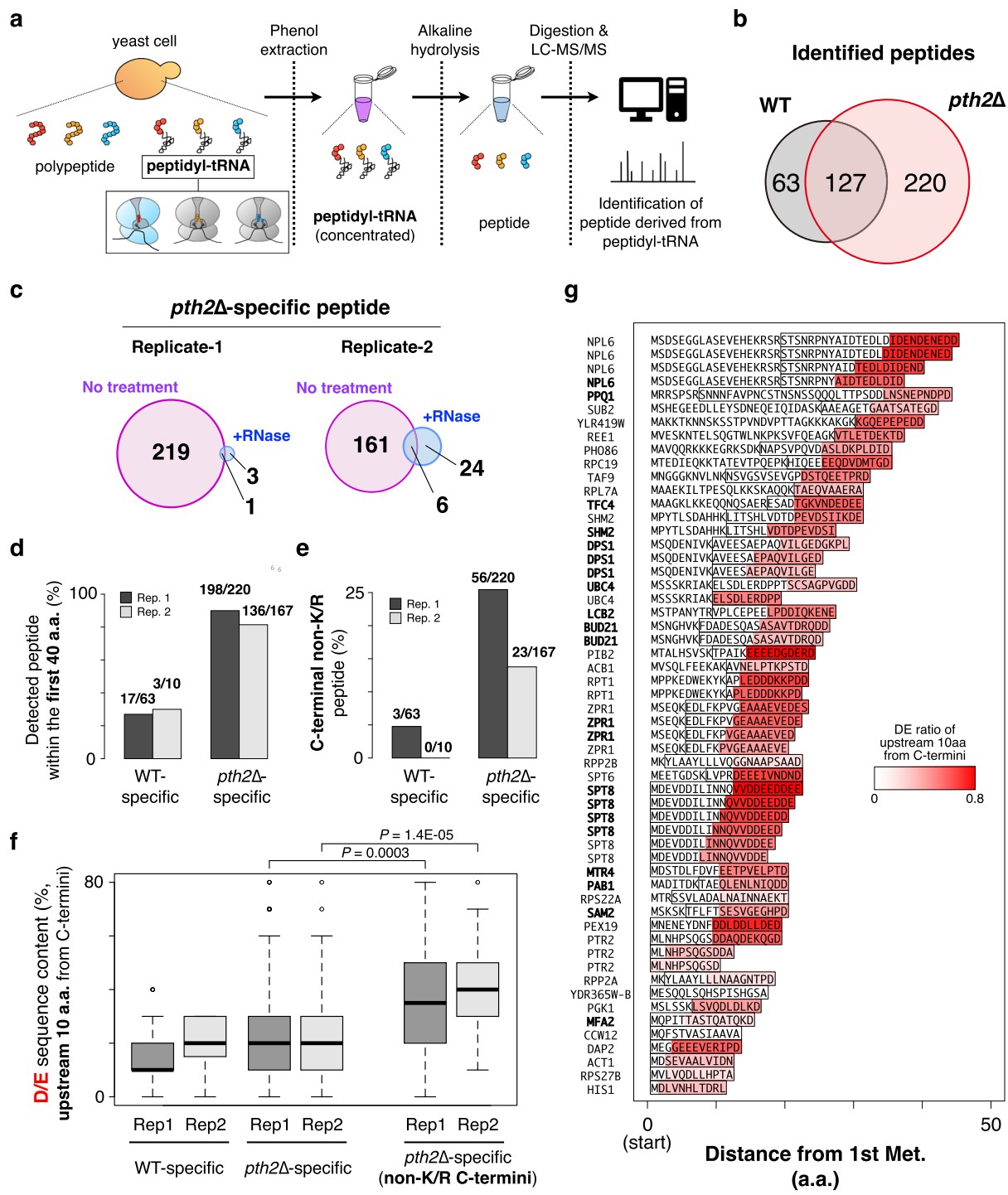

**a** yeast cell → Phenol extraction → Alkaline hydrolysis → Digestion & LC-MS/MS → Identification of peptide derived from peptidyl-tRNA

**b** Identified peptides — WT 63, 127, pth2Δ 220

**c** pth2Δ-specific peptide. Replicate-1: No treatment 219, +RNase 3, 1. Replicate-2: No treatment 161, +RNase 24, 6

**d** Detected peptide within the first 40 a.a. (%): WT-specific 17/63, 3/10; pth2Δ-specific 198/220, 136/167

**e** C-terminal non-K/R peptide (%): WT-specific 3/63, 0/10; pth2Δ-specific 56/220, 23/167

**f** D/E sequence content (%, upstream 10 a.a. from C-termini): P = 1.4E-05, P = 0.0003

**g** Distance from 1st Met. (a.a.), DE ratio of upstream 10aa from C-termini

contrast, our analysis showed that IRD-mediated translation abortion in eukaryotes only occurs when the D/E-rich sequences are located at the N-terminal regions. Indeed, there are many internal consecutive runs of negatively charged residues in eukaryotic ORFs. The bioinformatic survey revealed that longer (≥10) D/E-runs appear more frequently in eukaryotes than bacteria (Supplementary Fig. 8a). In budding yeast, dozens of genes encode more than 20 consecutive D/E residues in the middle of their ORFs. The most extreme one is Vhs3p[45,46], which includes 56 D/E runs starting from the 606th amino acid. The continuous translations after the long D/E runs seen in Vhs3p and others were confirmed by the GWIPS-viz browser (as global aggregate)[47], a public web tool to visualize the ribosome distribution.

One notable difference in the molecular mechanisms is that the eukaryotic ribosomes translating D/E-runs maintain the 80 S complex in a state where translation is inactivated. However, such inactivated 80 S complex is unstable under the $Mg^{2+}$-depleted condition, indicating that the translation of the D/E runs destabilizes and impairs the eukaryotic ribosome, but does not necessarily split the complex under the in vitro conditions. Consistent with this assumption, D/E-runs-derived peptidyl-tRNAs within the inactivated 80 S complex can be cleaved by Pth2p. In contrast, the elongation-arresting peptidyl-tRNAs were poorly cleaved by Pth2p, indicating that Pth2p can hardly access the peptidyl-tRNA accommodated within the intact ribosome complex, as also observed in E. coli IRD[23]. From these observations, we conclude that the translation of

**Fig. 5 | Identification of the abortive peptidyl-tRNAs accumulated in the *pth2Δ* strain by LC-MS/MS analysis. a** Flowchart of the analysis of the abortive peptidyl-tRNAs in yeast cells. Peptidyl-tRNAs, which are enriched in the phenol-extracted RNA fraction from wild-type (WT) or *pth2Δ* strains, are chemically hydrolyzed and then digested by trypsin and LysC peptidase. The resultant peptides are identified by LC-MS/MS. **b** The numbers of peptides identified in each strain prepared from mid-log culture and their overlap are represented in the Venn diagram. **c** The numbers of *pth2Δ*-specific peptides prepared from mid-log cultured cells. The results of the two biological replicates are shown. To confirm that most of these peptides were derived from peptidyl-tRNAs, the numbers of peptides from the RNA fraction treated by RNase during the sample preparation and their overlap are represented in the Venn diagram. **d** The ratio of WT-specific and *pth2Δ*-specific peptides whose C-terminal position was within 40 amino acids from the first methionine. The results of the two biological replicates from mid-log cultured cells are shown. The numbers above the bar represent the number of all detected peptides and the peptides within the first 40 amino acids. **e** The ratio of WT-specific

and *pth2Δ*-specific peptides whose C-terminal amino acid was not Lys/Arg (C-terminal non-K/R). The results of the two biological replicates from mid-log cultured cells are shown. The numbers above the bar represent the number of all detected peptides and the C-terminal non-K/R peptides. **f** The distribution of the relative content of D/E residues in the upstream ten amino acids from the C-terminus of the detected peptides. The results of the two biological replicates from mid-log cultured cells are shown. The box portions and the central bands are described according to the 25th percentile and the median, respectively. The whiskers indicate the maximum and minimum values excluding outliers defined by 1.5 times the length of the box. *P*-value was obtained by Wilcoxon's rank sum test (two-sided). **g** Mapping of the identified C-terminal non-K/R peptides specific to the *pth2Δ* strain prepared from mid-log cultured cells. Box represents the identified peptides by LC-MS/MS. The relative contents of D/E residues in the upstream ten amino acids from the C-terminus of the detected peptides are shown as a red gradient. A bold letter in the gene name indicates that the corresponding peptide was identified from both two biological replicates.

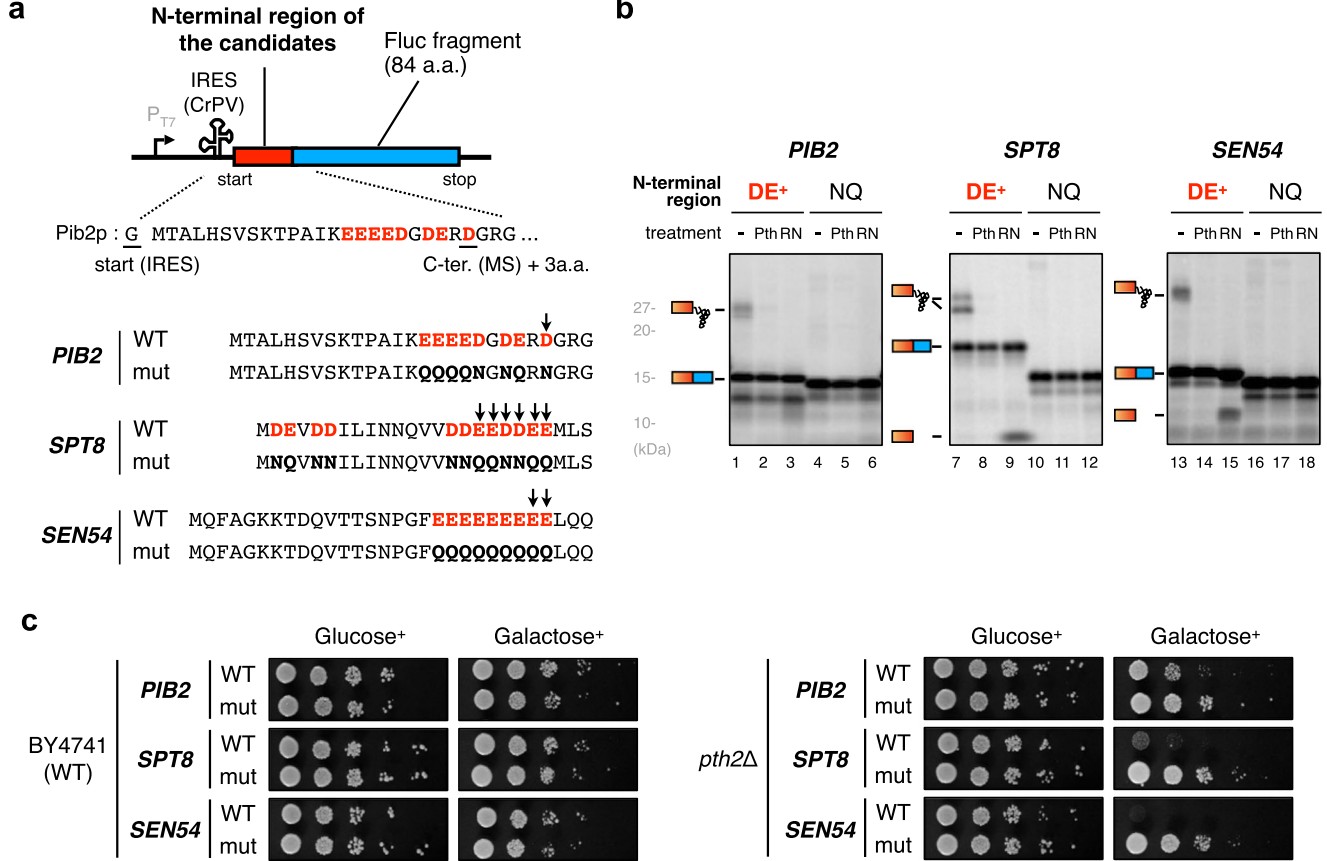

**Fig. 6 | IRD during the translation of LC-MS/MS-identified endogenous genes. a** Schematic representation of the sequences from endogenous genes for the HsPURE in vitro translation assay and the spot assay. N-terminal amino acid motifs detected by MS in *pth2Δ* cells (with 3 a.a. extension) were substituted for GFP_n−10X of the constructs in Fig. 1a and Fig. 1d. **b** IRD during the translation of endogenous genes depending on the D/E-rich sequences in the vicinity of the N-terminal region.

The reporter genes shown in Fig. 6a were translated using the HsPURE system, as described above. Representatives of two technical replicates are shown. **c** Toxicity upon the expression of endogenous IRD-inducing motifs. Wild-type (WT) and *pth2Δ* strains harboring plasmids expressing Fluc fused with the N-terminal sequence of the candidates were spotted on SD and SG plates lacking uracil and grown at 30 °C for 2–3 days.

D/E-runs somehow alters the conformation of the 80 S complexes to a suspended state in an IRD-dependent manner, and Pth2p rescues such impaired complexes. Recently, molecular simulations by Leininger and colleagues indicated that the translation of consecutive negative charges could shift the location of the nascent peptidyl-tRNA within the ribosome[48]. We speculate that this distortion of the peptidyl-tRNA or the 80S complex enables Pth2p to cleave the peptidyl-tRNA within. The prokaryotic ribosome complex may not be able to withstand such distortion and is thus prone to split, whereas the eukaryotic ribosomes may maintain the "80S" complex, probably

due to sophisticated peripheral bridges and the tunnel structure[49]. These questions should be addressed in future studies, including the structural analysis of the inactivated 80S complex.

A possible explanation for the presence of long D/E-rich sequences could be an extension of the functions of proteins using the regions in eukaryotes. The Gene Ontology (GO) analysis revealed that the human genes encoding proteins classified as being localized at the nucleus, chromosomes, and nucleolus, or binding to nucleic acids (DNA, RNA, histone, and chromatin), are enriched in D/E-runs (Supplementary Fig. 8b), as pointed out previously[50–52]. This suggests that the D/E-rich

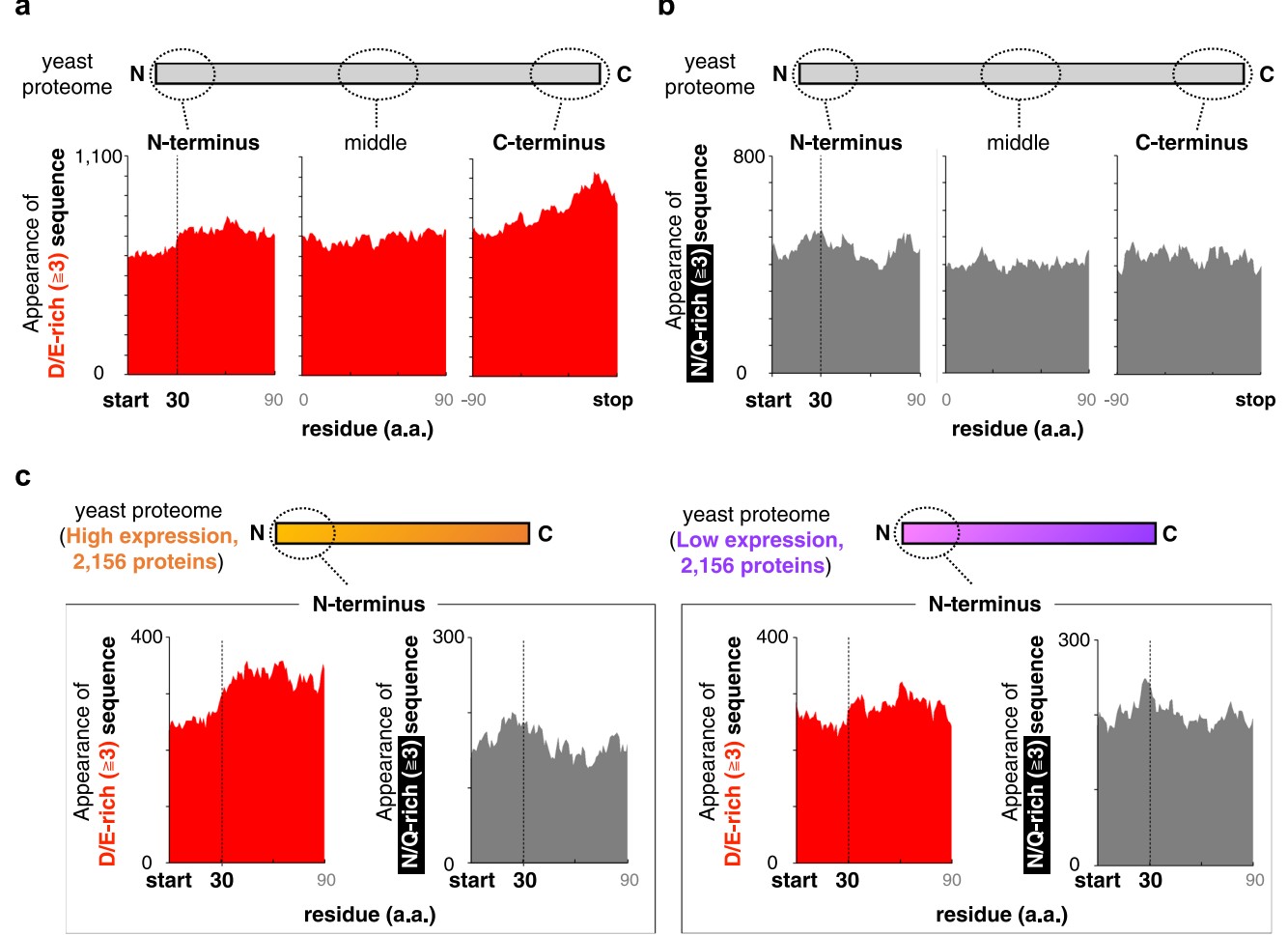

**Fig. 7 | Yeast proteome avoids positioning D/E-rich sequences in the N-terminal regions. a, b** Amino acid composition of the yeast proteome. The vertical axis represents the number of proteins harboring ≥3 D/E (**a**) or N/Q (**b**) in a 10 residue-moving window. N-terminus, C-terminus, and middle region are defined as N-terminal 100 amino acids excluding the first methionine, C-terminal 100 amino acids, and middle 100 amino acids of the 5,540 yeast proteins that have lengths greater than 130 amino acids. **c** Amino acid composition of the yeast proteome, depending on the protein abundance. The abundance data were obtained from the literature[44].

sequences are important for the acquisition of nuclear functions in eukaryotes. Domains including D/E-rich sequences can be regarded as "low (sequence) complexity" domains, which have recently received increasing attention as mediators to regulate liquid-liquid phase separation in the cell[53,54]. Indeed, the negatively charged Nephrin intracellular domain forms phase-separated nuclear bodies[55]. Other studies also indicated that "hyperacidic" sequences are utilized as interacting domains or interaction regulators for various proteins[45,46,56–59]. In addition, essential yeast proteins, such as the RNA binding protein Nab3p[60,61], the transcription elongation regulator Spt5p[62,63], and the 40S subunit biogenesis protein Kri1p[64,65], contain 23, 19, and 15 D/E runs from the 114th, 148th, and 52nd residues within the 802, 1063, and 591 aa length polypeptides, respectively. Collectively, stronger IRD counteraction would be reinforced to ensure the robust continuation of translation elongation for more complex eukaryotic proteomes.

Eukaryotic ribosomes are strengthened by several mechanisms, including eukaryote-specific intersubunit bridges such as eL19 and eL24 in the peripheral regions[49,66]. These features would make the eukaryotic ribosome association more sturdy for the translation of D/E-rich sequences, as compared to the association of prokaryotic ribosomes. In addition, a recent comparative analysis of exit tunnel structures revealed that the tunnel in eukaryotes has a narrower part and a second constriction site[2], suggesting a stronger interaction between the nascent chain and the interior of the tunnel. Our recent analysis in *E. coli* revealed

the nascent chain "length"-dependent IRD counteraction at the early stage of translation, and we proposed that the nascent peptide chain in the exit tunnel has a built-in ability to ensure elongation continuity by serving as a "bridge" with the interior of the tunnel[22]. Eukaryotic translation systems also show frequent IRD at the N-terminal regions and the "length"-dependent IRD counteraction, consistent with the properties of the bacterial IRD. Therefore, we assume that the narrower eukaryotic tunnel would provide a better chance for stabilizing and be a prerequisite to synthesize the "high-risk" sequences.

Recent studies revealed that eIF5A (Hyp2p) alleviates stalling on many motifs, including Asp-rich motifs such as DDG, besides the previously known proline stretches[18,19], indicating that the negatively charged residues during elongation could also be problematic. An in vivo reporter analysis using the *hyp2* temperature-sensitive strain showed unaltered IRD-induced premature termination in the eIF5A-impaired cells, demonstrating that eIF5A is unlikely to rescue the defective ribosome in the IRD state, since EF-P, the bacterial homolog of eIF5A, is not associated with IRD in *E. coli*[23]. It is currently unclear why the DDG motif and consecutive D/E residues behave differently. Further studies, including detailed structural analyses, will be necessary to elucidate the differences between the two phenomena.

We found that only Pth2p is involved in the cleavage of polypeptidyl-tRNAs derived from IRD in yeast cells. *S. cerevisiae* possesses two peptidyl-tRNA hydrolases, Pth1p and Pth2p, orthologous to

those in bacteria and archaea, respectively[36]. Previous comprehensive analyses annotated both Pths as localized to mitochondria[67,68]. However, Ishii et al. reported an interaction between Pth2p and the cytosolic Rad23-Dsk2 proteins that shuttle ubiquitinated proteins to the proteasome[69]. Thus, we assume that Pth2p exists "on" the cytosolic side of the mitochondrial membrane to cleave the peptidyl-tRNA in the cytosol. The observations that *E. coli* Pth rescues the lethal phenotype of *pth2Δ* upon the expression of D/E-runs and the association of the polyacidic peptidyl-tRNA with the 80S ribosome complex support our assumption. However, more careful analyses on the localization of Pth2p should be done in the future. With regard to the physiological perspective, in addition to the regulation of ubiquitinated protein degradation, Pth2p might associate with the clearance of the abortive peptidyl-tRNAs. The Pth2p-dependent inhibition of the substrate recruitment to the proteasome by Rad23-Dsk2 raises the interesting possibility that the IRD-derived peptides resolved by Pth2p could escape from degradation.

We developed an MS-based method to identify the endogenous peptidyl-tRNAs that are specific to cells lacking Pth2p. The properties of the identified peptidyl-tRNAs strongly suggested that IRD produces the peptidyl-tRNAs that are cleaved by Pth2p in wild-type cells. Although we identified ~220 *pth2Δ*-specific peptidyl-tRNAs by this method, we note that the scale of such premature products is probably underestimated. The peptidyl-tRNA species concentrated by the phenol extraction would be biased and contain some nonspecific contaminants. Also, the comprehensive identification of the entire peptidyl-tRNA-derived peptides is hampered by the technical limitations of MS, such as low ionization of the peptides and insufficient sensitivity. Despite these caveats, our new method clearly demonstrated that the prematurely terminated peptidyl-tRNA species are generated in an IRD-dependent manner, and are hidden in wild-type cells.

In conclusion, our results definitively demonstrated that both prokaryotic and eukaryotic ribosomes encounter some difficulties in synthesizing consecutive negatively charged amino acid sequences. In terms of the continuation of elongation, IRD might be regarded as a defect in translation, but *E. coli* conversely harnesses this weakness of the translational machinery as an environmental sensor[23]. Although the physiological relevance of IRD in eukaryotes remains enigmatic, it is possible that IRD could be utilized for cellular functions such as gene expression regulation or expansion of polypeptide repertoires. The bioinformatics analysis revealed that most of the proteomes in all kingdoms of life tend to avoid D/E-rich sequences at the N-terminal position (Supplementary Fig. 7). This tendency is consistent with the previous reports for eukaryotes[16,70] and prokaryotes[22], and is more obvious for genes with higher expression levels. Although this is reasonable for efficient protein synthesis, it should be noted that many other eukaryotic genes maintain the "risky" sequences that could induce translation abortion, perhaps reflecting the demands for IRD to regulate cellular functions. Further studies are required to clarify the biological significance of IRD in eukaryotes.

## Methods
### Construction of yeast strains and plasmids
Yeast strains, plasmids, and oligonucleotides used in this study are listed in Supplementary Data 2, 3, and 4, respectively.

For gene deletions, a cassette containing only a selection marker was PCR amplified as described[71,72], and introduced into the BY4741 background strain. All *S. cerevisiae* strains and PCR-amplified DNA fragments used for gene disruption in this study are summarized in Supplementary Data 2.

Plasmids were constructed using standard cloning procedures and Gibson assembly[73]. Detailed schemes are summarized in Supplementary Data 3, and the sequences of constructed plasmids are available in the Mendeley repository.

### Yeast cultures
Yeast cells were cultivated at 30 °C in either liquid yeast extract–peptone–dextrose (YPD)-rich medium, in synthetic dextrose, galactose or raffinose based on casamino acids (SD/CA, SG/CA or SR/CA) minimal medium (6.7 g/L yeast nitrogen base without ammonium sulfate, with 5.0 g/L casamino acids and 2% glucose), or in synthetic dextrose, galactose or raffinose based on dropout (SD/DO, SG/DO or SR/DO) minimal medium (6.7 g/L yeast nitrogen base without ammonium sulfate with a dropout solution for each amino acid). Yeasts harboring plasmids for luciferase assays were grown in SD or SG lacking uracil; and yeasts harboring plasmids for the overexpression of yeast Pth1p, Pth2p, or *E. coli* Pth were grown in SD or SG lacking leucine.

### Dual-luciferase reporter assay
Dual-luciferase assays were performed with the dual-luciferase reporter assay system (Promega Corp., Madison, WI). All reagents were prepared according to the manufacturer's instructions. *S. cerevisiae* strains were grown to exponential phase in SR/CA medium. The cells were harvested by centrifugation and resuspended in SG/CA. After induction for 4 h, a 10 μL portion of the culture was transferred to 100 μL of 1× passive lysis buffer. After allowing lysis to progress for 10 sec, a 10 μL aliquot was used for luminescence measurements with a Varioskan LUX Multimode Microplate Reader (Thermo Fisher Scientific). The following steps were performed on 96-well plates. A 50 μL portion of the firefly luciferase reagent (LARII) was added to the plate, with a 10 sec equilibration time and measurement of luminescence with a 10 sec integration time, followed by the addition of 50 μL of the *Renilla* luciferase reagent and firefly luciferase quenching (Stop & Glo), a 10 sec equilibration time, and measurement of luminescence with a 10 sec integration time. The data are represented as the ratio of Firefly to *Renilla* luciferase activity (Fluc activity / Rluc activity).

### Spot assay
*S. cerevisiae* strains were grown to exponential phase in appropriate media. The cultures were then serially diluted ($10^7$, $10^6$, $10^5$, $10^4$, and $10^3$ cells/mL), and 4 μL portions of the diluted cell suspensions were spotted on appropriate plates and incubated at 30 °C for 2–3 days.

### RNA isolation for peptidyl-tRNA analysis
*S. cerevisiae* strains were grown to exponential phase in SR/CA medium. The cells were harvested by centrifugation and resuspended in SG/CA. Total RNA was isolated using the TriPURE isolation reagent (Sigma-Aldrich) according to the manufacturer's instructions.

### In vitro peptidyl-tRNA hydrolysis
The RNA sample was divided into four portions. Two were resuspended in HEPES-KOH (pH 7.6) and incubated with 1 μM of Pth2p at 30 °C for 20 min or 1 μM of Pth from *E. coli* at 37 °C for 20 min, and the others were resuspended in 100 mM Tris (pH 11.0) followed by heating at 80 °C for 20 min.

### Northern blot
RNA samples separated by 11% WIDE Range Gel SDS-PAGE were transferred onto a Hybond-N⁺ membrane (GE Healthcare) and hybridized with biotinylated oligonucleotide(s) (IDT) complementary to the tRNAs shown below with the probe sequences: tRNA^Asp: CCGCGACGG GGAATTGAACCCCGATCTG, tRNA^Asn: CCCCAGTGAGGGTTGAACTCAC GATCTT, 5SrRNA: ACCCACTACACTACTCGGTCAGGCTCTTAC. Hybridization experiments were performed using a NorthernMax kit (Ambion) and a Chemiluminescent Nucleic Acid Detection Module (Thermo Scientific) according to the manufacturer's instructions. Images were visualized and analyzed by an LAS4000 LuminoImager (GE Healthcare).

## Sample preparation for LC-MS/MS analysis

For the identification of the peptides from the GFP$_5$-(X)D peptidyl-tRNAs, RNA samples isolated by the TriPURE reagent and isopropanol precipitation were resuspended in PTS buffer, consisting of 12 mM sodium deoxycholate (SDC), 12 mM sodium *N*-lauroyl sarcosinate (SLS), and 100 mM Tris (pH ~11). The solution was then incubated at 80 °C for 20 min to hydrolyze the peptidyl-tRNA. After the hydrolysis, PTS buffer consisting of 12 mM SDC, 12 mM SLS, and 100 mM Tris-HCl (pH 6.8) was added to adjust the pH to 9.

The obtained protein/peptide solution was reduced with 10 mM DTT for 30 min at room temperature, and the resultant proteins/peptides were alkylated with 50 mM iodoacetamide for 30 min at room temperature in the dark. After a four-fold dilution with 50 mM ammonium bicarbonate, the proteins/peptides were digested with endoproteinase Lys-C (Wako Pure Chemical) at room temperature for 3 h. Proteins/peptides were further digested with trypsin (Trypsin Gold, Promega) at 37 °C overnight. The amounts of added Lys-C or trypsin were 1 μg per 100 μg total protein or 1 μg per 50 μg total protein, respectively. After the digestion, SDC and SLS were removed by ethyl acetate extraction at low pH (0.5% trifluoroacetic acid). The samples were then desalted with a desalting column (GL-Tip SDB, GL Sciences) and dissolved in a 2% acetonitrile and 0.1% trifluoroacetic acid solution before the LC-MS/MS measurement.

For the identification of the endogenous substrates of Pth2p, RNA samples were isolated from yeast cells cultured in 200 mL of YPD medium at the mid-log phase or at the late-log phase by the TRIZOL reagent (Invitrogen) and isopropanol precipitation. Isolated RNA and peptidyl-tRNA were resuspended in PTS buffer-A (12 mM SDC, 12 mM SLS, and 100 mM Tris-HCl pH 6.8). The solution was then incubated at 37 °C for 30 min with shaking, then centrifuged to remove the insoluble precipitates. The collected supernatant was divided into two portions and half of them was treated with a final 100 μg/mL of RNase A (Promega) at 37 °C for 60 min. Samples were again extracted with an equal volume of TRIZOL LS reagent (Invitrogen) and following isopropanol precipitation. The precipitate was resolved in PTS buffer-B (2.4 mM SDC, 2.4 mM SLS, and 100 mM Tris-HCl pH 11) and was then incubated at 80 °C for 20 min to hydrolyze the peptidyl-tRNA. After the hydrolysis, 1/5 volume of 1 M Tris-HCl (pH 6.8) was added to adjust the pH to 9.

The obtained peptides derived from peptidyl-tRNAs were reduced with 10 mM DTT for 30 min at room temperature, and the resultant proteins/peptides were alkylated with 50 mM iodoacetamide for 30 min at room temperature in the dark. Then samples were digested with a final 0.5 μg/μL of trypsin/Lys-C mix (Trypsin Gold, Promega) at 37 °C for 3 h. An equal amount of trypsin/Lys-C mix was again added and the peptides were further digested at 37 °C overnight. After the digestion, SDC and SLS were removed by ethyl acetate extraction at low pH (0.5% trifluoroacetic acid). The samples were then desalted with a StageTip composed of an SDB-XC Empore disk (3 M, U.S.A.) and dissolved in a 2% acetonitrile and 0.1% trifluoroacetic acid solution before the LC-MS/MS measurement.

## LC-MS/MS analysis

LC-MS/MS measurements were performed with a nanoLC-ESI-MS/MS system composed of a quadrupole-orbitrap hybrid mass spectrometer (Q-Exactive; Thermo Fisher Scientific) equipped with a nanospray ion source and a nano HPLC system (Easy-nLC 1000; Thermo Fisher Scientific). The trap column used for the nano HPLC was a 2 cm × 75 μm capillary column packed with 3 μm C18-silica particles (Thermo Fisher Scientific) and the separation column was a 12.5 cm × 75 μm capillary column packed with 3 μm C18-silica particles (Nikkyo Technos Co., Ltd.). The flow rate of the nano HPLC was 300 nL/min. The separation was conducted using a 10–40% linear acetonitrile gradient for 30 min for the identification of IRD candidates (GFP$_5$-(X)D peptides and late-log, replicate 1) or 70 min for the identification of IRD candidates (mid-log, replicate 1 and 2, and late-log, replicate 2) and the quantification of

luciferase peptides in the presence of 0.1% formic acid. Each sample was measured three times as technical replicates. The LC-MS/MS data were acquired in the data-dependent acquisition mode controlled by Xcalibur 4.0 (Thermo Fisher Scientific). The settings of the data-dependent acquisition were as follows: the resolutions were 70,000 for the full MS scan and 17,500 for the MS2 scan; the AGC targets were 3.0E6 for the full MS scan and 5.0E5 for the MS2 scan; the maximum IT was 60 msec for both the full MS and MS2 scans; the scan range was 310–1500 *m/z* for the full MS scan, and the top 10 signals were selected for the MS2 scan per one full MS scan, and the dynamic exclusion was 15 sec.

For the evaluation of GFP$_5$-(X)D peptides, extracted ion chromatograms of the corresponding peptide and isotope dot product (idotp) scores were obtained with the Skyline software[74].

For the identification of the peptides derived from peptidyl-tRNAs, the Proteome Discoverer 2.4 software (Thermo Fisher Scientific) bundled with the Sequest HT search engine was used. The MS/MS spectra were searched against all *S. cerevisiae* ORF sequences obtained from the *Saccharomyces* Genome Database[75] (https://www.yeastgenome.org/, downloaded on December 24, 2019; last modified on January 13, 2015) by using a semi-tryptic digestion search setting. Other detailed settings of the search with Proteome Discoverer 2.4 were listed in Supplementary Data 5. Only the peptides with high false discovery rate (FDR) confidence (<0.01, by Percolator algorithm) and ≥3 number of peptide-spectrum matches (PSMs) were defined as identified peptides. The analyses of the identified peptides data (amino acid position, Asp/Glu sequence content, statistical analyses, etc.) were performed by the R software (for Mac, version 4.1.3) with in-house scripts.

## In vitro translation and product analysis

The coupled transcription-translation reaction was performed using the HsPURE system[29] in the presence of $^{35}$S-methionine, at 32 °C for 90 min. The coupled transcription-translation reaction using the *E. coli* PURE system was performed by using PURE*frex* (v1.0) in the presence of $^{35}$S-methionine at 37 °C for 30 min[22,23]. Template DNAs for in vitro transcription-translation reactions of the mRNA encoding CrPV IRES were amplified by PCR, using PM0771(GGCCTAATACGACTCACTATAG GGAAAAAGC) and PM0773(GTTATTGCTCAGCGGTTAAGAGTTTTCAC TGCATACGACG). In the case of the plasmids carrying HCV IRES, the template DNA was amplified by PCR using PM0511 and the GAL1_seq-primer (GCATAACCACTTTAACTAATACTTTCA). The template DNA for the *E. coli* PURE system was amplified by primer 1 (GGCCTAAT ACGACTCACTATAGGAGAAATCATAAAAAATTTATTTGCTTTGTGAGC GG) and primer 3 (AGTCAGTCACGATGAATTCCCCTAGCTTGG).

The reaction mixture of the HsPURE system was treated with 200 μg/mL of puromycin for 5 min at 32 °C if indicated. Subsequently, the reaction mixture was divided into two portions, and one was incubated with 1 μM of Pth2p at 32 °C for 20 min. The reaction mixture of *E. coli* PURE system was treated with 1 μM of purified Pth for 10 min at 37 °C when indicated. The reaction was stopped by dilution into an excessive volume of 5 % TCA. After standing on ice for at least 10 min, the samples were centrifuged for 3 min at 4 °C, and the supernatant was removed by aspiration. The precipitates were then vortexed with 0.9 mL of acetone, centrifuged again, and dissolved in SDS sample buffer (62.5 mM Tris-HCl, pH 6.8, 2% SDS, 10% glycerol, 50 mM DTT) that had been treated with RNasecure (Ambion). Finally, the sample was divided into two portions. One was incubated with 50 μg/mL of RNase A (Promega) at 37 °C for 30 min, and separated by the WIDE range SDS-PAGE system (Nacalai Tesque). The translation continuation index (TC index) was calculated by the following formula: {full-length polypeptide (10X)} / {full-length polypeptide (10X) + peptidyl-tRNA (10X)}.

## Sucrose gradient centrifugation of in vitro translation reaction mixture

The reaction mixture (10 μL at 32 °C, 60 min) of the HsPURE system was mixed with 100 μL of TNM buffer (25 mM Tris-HCl pH 7.6, 100 mM

$NH_4Cl$, 10 mM $MgCl_2$, 1 mM DTT), and centrifuged at $20,000 \times g$ for 10 min at 4 °C. Supernatant was recovered, layered on the top of 10–30% sucrose gradient containing TNM buffer in Open Top Poly-clear™ Centrifuge Tubes (14 × 89 mm, SETON) and centrifuged at 39,000 rpm for 2.5 h at 4 °C (Beckman OptimaL-90K, SW41-Ti), followed by fractionation (each fraction, ~350 μL) using Gradient Station (BIOCOMP) equipped with MICRO COLLECTOR AC-5700 (ATTO). Distribution of ribosomes was monitored by $A_{254}$ measurements using BIO MINI UV MONITOR (AC-5200S, ATTO). Each fraction was mixed with an equal volume of 10% TCA and processed for 11% WIDE Range Gel SDS-PAGE as described above. Translation products were visualized by northern blot analyses using 5′-biotinylated oligonucleotides (mixture of CCTGTTGGGGGACTCAAACTCCAGTCTCC, CCCGTCGGGGAGAGCGAA CCCCGCTCTCC, CCCGTTGGGGAATCGAACCCCGGTCTCC, CTCCTCA GGGAATCGAACCCCAGTCTCC, CCCGTCGGGGAATCGAACCCCGGTCT CC, CCCGTCGGGGAATTGAACCCCGGTCTCC) as described above.

## Toeprint analysis

Template DNAs for toeprint analysis were amplified by PCR, using PM0771 and PY0288(AACGACGGCCAGTGAATCCGTAATCATGGT-TATTATTATGTAAAAGCAATTGTTCCAG). In vitro translation reaction sample (32 °C, 40 min) was mixed with an equal volume of reverse transcription mixture {50 mM HEPES-KOH pH 7.6, 100 mM potassium glutamate, 2 mM spermidine, 13 mM magnesium acetate, 1 mM DTT, 2 μM fluorescently labeled oligonucleotide (pe-lacZ-N-rv with Alexa 647 at 5′-terminus), 50 μM of each of dNTPs and 10 unit/μL ReverTra Ace (Toyobo)} and incubated at 32 °C for 20 min. The reverse transcription products were purified by NucleoSpin Gel and PCR clean-up kit equilibrated with NTC buffer (Macherey-Nagel). Dideoxy DNA sequencing samples were prepared using Thermo Sequenase Primer Cycle Sequencing Kit (GE healthcare) and the same templates and primer (pe-lacZ-N-rv) as used for toeprint analysis. Samples were subjected to 6% polyacrylamide-7 M urea-TBE gel electrophoresis. Fluorescent images were visualized and analyzed by Amersham™ Typhoon™ scanner RGB system (GE healthcare) using 635 nm excitation laser and LPR emission filter. Ten mM of harringtonine was added beforehand, or the reaction mixture was treated with final 200 μg/mL puromycin or 1 μM of Pth2p for 20 min at 32 °C if indicated.

## Chase experiment

The HsPURE system reaction mixture (32 °C, 60 min, in the presence of $^{35}$S-methionine) was treated with an excessive amount of unlabeled methionine (final 2 mg/mL) to stop the incorporation of $^{35}$S-methionine into the translational products. Then the reaction mixture was further incubated at 32 °C and a portion was withdrawn and mixed with an excessive volume of 5% TCA at the indicated time points (0, 30, 60, 120, and 180 min after the addition of cold methionine).

## Amino acid composition analysis and enrichment analysis with Gene Ontology

The amino acid composition analysis was conducted with in-house R scripts (with R.app ver. 3.1.2). We defined the N-terminus, C-terminus, MID, and random regions as the N-terminal 100 amino acids excluding the first methionine, C-terminal 100 amino acids, and middle 100 amino acids of the 5446 yeast proteins with lengths greater than 130 amino acids, and used these sequences to calculate the amino acid composition. Amino acid enrichment was calculated using a ten-residue-moving window, and we counted proteins harboring more target amino acids than the determined number in a window. The same analysis was conducted for other organisms, as shown in Supplementary Fig. 7. All amino acid sequences were obtained from the UniProt database.

The enrichment analysis with Gene Ontology annotation for human proteome was conducted with the PANTHER web-tool (http://www.pantherdb.org/)[76].

## Reporting summary

Further information on research design is available in the Nature Portfolio Reporting Summary linked to this article.

## Data availability

The mass spectrometry data generated in this study are available at ProteomeXchange via the jPOST repository with the dataset identifier PXD031958. Sequences of constructed plasmids, raw data files are available in the Mendeley Data repository (https://data.mendeley.com/datasets/gcsvcsbs25/1). All other relevant data and materials are available from the corresponding author(s). Source data are provided with this paper.

## Code availability

Custom R codes for the mass spectrometry data analyses are provided as Source data and in the Mendeley Data repository (https://data.mendeley.com/datasets/gcsvcsbs25/1).

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

## Acknowledgements
We thank Kosuke Ito for providing Pth2p; Takashi Kanamori for providing *E. coli* Pth; Motonori Ota for valuable discussions; Eri Uemura for technical support; the Bio-support Center at Tokyo Tech for DNA sequencing; and the Cell Biology Center Research Core Facility at Tokyo Tech for the Q-Exactive mass spectrometry measurement. This work was supported by MEXT Grants-in-Aid for Scientific Research (Grant Numbers JP26116002, JP18H03984, JP20H05925 to H.T., 17K15062, 19K16038 to Y.C.) and a grant from the Ohsumi Frontier Science Foundation to Y.C.

## Author contributions
Y.I., Y.C., T.N., and K.M. performed experiments; Y.I., Y.C., T.N., A.Y., K.M., H.I., and H.T. conceived the study, designed experiments, and analyzed the results; Y.C. and H.T. supervised the entire project; Y.I., Y.C., and H.T. wrote the manuscript.

## Competing interests
The authors declare no competing interests.
