## [Peer Review File · Nature Communications]

Nascent peptide-induced translation discontinuation in eukaryotes impacts biased amino acid usage in proteomesREVIEWER COMMENTS

Reviewer #1 (Remarks to the Author):

Previous work from the same group identified a phenomenon in bacteria whereby translation of polyacidic sequences causes premature peptidyl-tRNA release. This process was termed intrinsic ribosome destabilization or IRD for short. The authors proposed that the polyacidic sequences caused a destabilization of the ribosome that was counteracted to some extent by ribosomal protein L31 as well as by increasing the magnesium concentration.

The manuscript of Ito et al addresses the question as to whether this process also occurs in eukaryotes, which is a reasonable assumption given the conservation of the translational apparatus. Using a series of in vivo and in vitro assays the authors present convincing evidence that this is indeed the case and identify a number of endogenous proteins that undergo IRD in yeast. Overall, I find the manuscript to be very nicely written and presented. As far as my experience allows (i.e. I am not a mass spectrometry expert), the experiments appear to be technically well performed and appropriately interpreted and the results support the conclusions. In my opinion the main weakness of the manuscript (and perhaps also for the previous bacterial version) is related to the mechanism and order of events (see points below). Any additions in this direction would certainly strengthen the manuscript.

1. I am still a little confused by the order of events in this IRD process, even in relation to the bacterial system. Does the polyacidic peptidyl-tRNA cause subunit splitting (which is the destabilization of the ribosome that is always talked about?) and this is the cause of the peptidyl-tRNA drop-off? If the authors want to use the term IRD for eukaryotes then shouldn't they actually show that subunit splitting by polyacidic peptidyl-tRNA is also occurring in the eukaryotic system?
2. In relation to point 1, are there actually experiments showing that the peptidyl-tRNA is actually dissociating from the ribosome and this is the substrate for the PTH? I don't see any experiments excluding the possibility that the PTH can actually cleave the peptidyl-tRNA while its still bound to the large subunit (but after splitting). The manuscript lacks any summary model schematic figure...the previous bacterial paper had a summary only in the graphical abstract which suggested splitting but was very vague in terms of the order of events.
3. The bias against polyacidic sequences in the N-terminal 30aa appears relatively modest. One wonders why these sequences are not simply selected against if they induce IRD i.e. are these sequences actually conserved in these proteins across different organisms (eukaryotes, but maybe archaea and bacteria too?). Somehow this reminds me of the polyproline situation except here life evolved a solution e.g. EF-P and eIF5A so as to be able to maintain the presence of polyproline sequences in proteins. But here this is not the case. If I remember correctly, there was only one protein Val-tRNA synthetase that has a conserved PPP motif across all kingdoms of life. How is it for polyacidic sequences?

Reviewer #2 (Remarks to the Author):

Comments to the authors:

In their manuscript Nascent peptide-induced translation discontinuation in eukaryotes impacts biased amino acid usage in proteomes, Ito, Chadani and colleagues from the Taguchi lab demonstrate for the first time that N-terminal enrichment in aspartic acid and glutamic acid residues can lead to intrinsic ribosome destabilization, not only in bacteria, where the group had discovered this phenomenon, but also in eukaryotes. They find that this happens only at N-termini but not at internal positions, in contrast to bacteria, suggesting the eukaryotic ribosome is less prone to the phenomenon. They also show that a particular peptidyl-tRNA hydrolase, Pth2 in yeast, cleaves peptidyl-tRNAs that result from this effect. Upon deletion of Pth2, they found enrichment of peptidyl-tRNAs of which many

were enriched in glutamic acid and aspartic acid residues.

They finally show that stretches of D/E residues are disfavored at the first 30-40 positions in the yeast proteome, especially in highly expressed proteins.

This manuscript outlines an effect that has been discovered by the same group about 5 years ago and had not been shown to exist in eukaryotes before. In the light of recent great interest in translation, failure of translation, and translation quality control, I believe that this finding is of high interest to the wider community. The manuscript also introduces a novel methodology to identify peptides from peptidyl-tRNAs via mass spectrometry.

In my opinion, the experiments were performed to a high standard and support the messages well.

I found the manuscript well written, easy to follow, and enjoyed reading it. The methods are generally well described. The figures are well designed, as well.

Here are specific comments I have about the manuscript:

- 1) In Fig. 1 I found it difficult to compare TC indices between panel B and panel C due to the difference in representation (horizontal vs. vertical).
- 2) Why are the results different between Fig. 1F and 1G for the GFP5-10D without added Mg²⁺? (~0.18 vs. ~0.05) Am I correct that in Fig 1F 0 mM added Mg²⁺ was applied?
- 3) Would it make sense to speculate why RTF1 has a much lower TC than ANB1 and NMD2? Could it be because the all 4 D/Es are almost in a single stretch with only 1 other amino acid between?
- 4) Is there any hint, e.g., in the literature that endogenous IRD targets, such as JIP3 are upregulated upon high Mg²⁺ concentrations in the cell? To my knowledge this would occur, e.g., upon ATP depletion.
- 5) Are D/E rich N-termini of endogenous IRD targets conserved across species?
- 6) Fig. 4 Are non D/E ending peptides generally enriched in D/E above average?
- 7) Related to 6: Could the D/E rich C-terminus of peptidyl-tRNA peptides have been lost due to LysC cleavage and poor flyability? If you analyze the peptide composition following the K/R cleavage site, is there an enrichment in D/E?
- 8) I found the graph in Figure S6B hard to understand. I believe the curve should not be smoothed as there are 15 discrete steps from 0 to 15 D/E residues, so it should look like a step graph.
- 9) In Figure S7 the region in which D/E residues are disfavored appears to be a lot shorter than in yeast. I believe this is noteworthy and should be commented. It might also be very interesting to repeat the analysis for a few more different eukaryotes, to check if this is something that is, e.g., common to mammals or similar.
- 10) I was wondering which datasets were inspected in the GWIPS viz analysis. This made me realize that I did not find any description of this analysis in the Methods part. I believe it should be added.
- 11) In the description of how the rolling-window analyses of D/E richness was done, it is described that all genes above 130 amino acids length were separated into 100 amino acid N-terminus, 100 amino acid middle and 100 amino acid C-terminus. Should this not only work for genes of at least 300 amino acids length? What happened for those shorter? Were they excluded or were 3 overlapping regions included in the analysis?

12) In l. 280-282 it is written:

"Strikingly, we found that the number of proteins with ≥ 3 D/E residues in the N-terminal regions dropped early, from the N terminus to 30-40 amino acids, which is similar to the length of the tunnel." I believe that the term "dropped" is not the best to describe this, as this implies that the number of such proteins is high at first and then goes down. For what I see, this is not the case. A better term might be "is low", but there might be better ways to describe this. Consider re-phrasing this sentence.

13) l. 390-392:

"Second, the comprehensive identification of the entire peptidyl-tRNAs is hampered by technical limitations of MS, such as low ionization of the peptides and insufficient sensitivity." I would argue that it is not the entire peptidyl-tRNA but "peptidyl-tRNA derived peptide", as the tRNA is cleaved off prior to mass spectrometry. Therefore, one might consider rephrasing, possibly also at other points in the paragraph to ensure it is clear that only peptides are analysed, not the tRNA part.

14) l. 536-537:

"In this analysis, every ten peptides with high reproducibility were selected and used for comparison." I do not fully understand what is meant by this sentence. Can you explain a bit more?

15) Figure caption 1E) "Radioactive bands were detected by a phosphorimager." Technically they were surely detected with the help of storage phosphor screens?

16) It would be good to mention in the abstract that most of the study was done on *S. cerevisiae*.

17) Table S1: There is a strain named SCYXXX. I believe this has to be named.

Reviewer #3 (Remarks to the Author):

Taguchi and colleagues have previously determined that translation of an mRNA in bacteria is aborted due to intrinsic ribosome destabilization (IRD) if the open reading frame (ORF) contains consecutive codons at or near its 5'-end that encode negatively charged amino acid residues. Here, these researchers report the results of a logical and well-executed series of experiments showing that IRD also occurs in eukaryotes, and that the dependence of IRD on specific sequences near the N-terminus of polypeptides leads to selection against their occurrence in eukaryotic proteomes. The finding that the peptidyl-tRNA hydrolase Pth2p is involved in the cleavage of the abortive polypeptidyl-tRNAs generated by IRD suggest that they are released from destabilized ribosomes: the occurrence of IRD in a fully in vitro reconstituted human translation system that contained only ribosomes, tRNAs, elongation factors, termination factors and aminoacyl-tRNA synthetases excluded the involvement of known recycling factors in the splitting of ribosomes. However, splitting of arrested ribosomes into subunits was not shown, and the mechanism of IRD remains unknown. Although the authors do not explicitly state how they propose that nascent peptidyl-tRNAs are released from ribosomes following termination, line 387 suggests drop-off. Notably, the authors also showed that elevation of Mg concentrations alleviates IRD and allows uninterrupted translation of proteins containing N-terminal D/E-rich stretches. It would add to the report if the authors tested whether arrested ribosomes could resume translation after elevation of Mg concentrations and if they could, whether such an ability depends on the time of initial incubation at low Mg concentrations that are permissible for IRD. These experiments can be done in the in vitro reconstituted system and resumption of translation can be monitored by disappearance of peptidyl-tRNAs. The proposed experiments could potentially estimate how quickly presumable drop-off of peptidyl-tRNA occurs in the reconstituted system.

SPECIFIC COMMENTS

1. It would be helpful to readers if the authors explained why they included the CrPV IGR IRES in their mRNAs.

2. Typographical and other minor errors.

Line 135. It would be more accurate to state that additional Mg⁺⁺ partially alleviated IRD.

Line 154. The term 'immaturely' should be replaced by 'prematurely'.

Figure 1G. Why "+/-" 0?

Lines 184-185. Verma et al (Ref. 36) claimed that Vms1p is a peptidyl-tRNA hydrolase and not an endonuclease, so that this description is inaccurate.

Reviewer #4 (Remarks to the Author):

The manuscript "Nascent peptide-induced translation discontinuation in eukaryotes impacts biased amino acid usage in proteomes" by Ito and colleagues describes that E/D rich sequences lead to translation abortion and the release of peptidyl-tRNAs. The authors show that such abortion events preferentially occur in the N-terminal part of proteins introducing a fitness burden by toxic peptidyl-tRNAs that are subsequently neutralized by Pth2. The authors find that such sequences are underrepresented in the N-terminal part of proteins indicating that the associated negative selection pressure shaped the current yeast proteome.

The current manuscript is a continuation of the work of the Taguchi lab on destabilizing sequences, which was extended to translation in eukaryotes. The results are interesting and new and will contribute to a quickly developing field at the intersection of translation, QC events, protein folding and the coding capacity of the genetic code. As such, the manuscript will be of interest for a broad audience of readers. The manuscript is largely technically sound and central findings are supported by orthogonal methods in vivo, in vitro, and in silico. However, I have concerns about the conceptual sharpness of the mechanistic interpretation, the data presentation and the validation of the MS assay used.

Major points:

Conceptual clarity:

1. While the manuscript shows convincingly that translation is terminated by specific acid amino acid sequences and the respective peptidyl tRNAs accumulates in the cell, the exact mechanism of the abortion is not entirely clear. Peptidyl transfer of highly acidic sequences could be slowed down kinetically favoring the drop-off of the peptidyl-tRNA from the ribosome. Second, highly acidic nascent chains could destabilize the peptidyl-tRNA leading to their dissociation. Third, highly acidic nascent chains could destabilize the ribosome leading to IRD (intrinsic ribosome destabilization) events as proposed recently (Chadani et al., 2017). How do the authors discriminate these scenarios? What is the basis for the conclusion that highly acidic sequences destabilize the ribosome? Increasing Mg²⁺ concentrations (e.g. Fig1G & S1E) of course stabilize the subunit association but also many other RNA interactions such as the binding of the peptidyl-tRNA to the P site. How do the authors discriminate peptidyl-tRNA drop-off from the ribosome from an IRD event? Is there biochemical or biophysical evidence that ribosomes are indeed dissociated? How can the length effect of the nascent peptidyl chain can be interpreted in the light of IRD events? While it is easy to understand that long peptidyl-tRNA do not drop off from the ribosome, it is unclear why they should not dissociate the ribosome in IRD events (as seen in the E. coli system). As the motivation for the paper was to understand "whether IRD takes place even in eukaryotic cells" (line 57), in particular because the eukaryotic ribosome association appears to be more robust than the prokaryotic one (line 351), these controls have to be provided. Referring to the high conservation of ribosome dynamics among different organisms (line 316) and the high conservation of the ribosome exit tunnel (line 318) is not sufficient. The authors should sharpen their mechanistic conclusions and provide essential controls that show that these sequences indeed destabilize the ribosome itself.

2. The authors state that they find "many endogenous D/E-rich peptidyl-tRNAs... in cells lacking a peptidyl-tRNA hydrolase". However, in the corresponding Figure 4 and Figure S4 peptides are categorized based on their C-terminal amino acid. Using this definition AVNELPTKSTD (Figure 4E) is D/E rich while ADESDPVESK (Figure S4B) is not D/E rich. What is the rationale for this definition? If

the author would like to conclude that the PTH-dependent peptides are D/E rich, they should analyze the D/E enrichment as well.

Data presentation & validation: Figure 4

3. The MS data are difficult to evaluate. Deposited files are named Ito1, Ito2, etc. The submitted peptides and proteingroups files do not contain the data depicted in Figure 4. The parameters of the MaxQuant search are not comprehensively described in the Method section (FDR, variable modifications...). The authors should provide a more user-friendly, better annotated data access.

4. The enrichment of nascent chains by phenolisation and their subsequent mass spectrometric detection is elegant and relies on well-established analysis workflows. The enrichment of acidic sequences in Δ pth2 cells and their N-terminal location is convincing and in agreement with the translation experiments in vivo and in vitro. However, the analysis is qualitative only. It is unclear whether the experiment was replicated, as only data for one biological replicate was deposited. The nature of the technical replicates is unclear as well. The authors should state how the experiment was replicated. Given that the authors present the assay for the first time, replicates would help to judge the robustness of the results and the enrichment of peptidyl-tRNAs.

5. The authors state that they aimed to "broadly explore the ORFs that prematurely terminate the translation". The analysis revealed 36 target peptides originating from an even lower number of proteins. The authors conclude that the occurrence of drop-off products might be underestimated and that technical limitations such as low ionization of target peptides and insufficient sensitivity hamper the analysis (line 391). It is unlikely that in the analysis of yeast lysates the sensitivity of MS itself is limiting, because the amount of available sample should not be limiting and the authors could have loaded more peptide digest per analysis. Inspecting the RAW files indeed shows that the authors were able to saturate the LC-MS system. Rather, a combination of the only moderate enrichment of target peptides (all less than 100 intense compared with the most prominent peptides), the high concentration of lab contaminants and the limited dynamic range of MS presumably led to the limited number of hits. As the authors state, the discovery of the target peptides is likely impaired by their poor ionization. However, if a more comprehensive analysis of abortion products would be desired, biological replicates, prefractionation on the peptidyl-tRNA level (e.g. anion exchange separation) or on the peptide level (reverse phase or size exclusion), in-gas fractionation or simply longer LC runs (currently only 30 min gradients) could likely boost the depth of analysis. Because trypsin/LysC treatment leads to poorly ionizing semi tryptic peptides, LysN/ArgN or TrypsinN might be better options. They should produce better ionizing peptides and lead to cleaner MS/MS spectra due to more complete b-ion ladders. Both should result in a higher observability and better peptide scores. If the authors are satisfied with the current depth of analysis, they should better express the qualitative nature of the experiment. Alternatively, the authors might apply one of the commonly used strategies to overcome these technical issues. This would allow to better estimate how frequently these abortion events occur.

6. I assume that tRNAs connected to longer peptidyl-chains or more hydrophobic sequences are more likely located in the interphase after phenol extraction. How do the authors consider such potential bias towards the isolation of short, hydrophilic peptidyl-RNAs?

Minor points:

1. The authors should discuss their results in the light of the available literature on the effect of highly acidic sequences in the PTC (e.g. Leininger et al., 2021; PMID: 34652913).
2. Given that the cellular localization of Pth2 seems to be under debate, the authors should discuss the cellular localization of peptidyl-tRNAs that are enriched.
3. The bioinformatics analysis revealed that negatively charged amino acids are avoided in the N-terminal part of proteins. At the end of the discussion the authors mention that this is consistent with previous reports (line 403). Why was it necessary to reanalyze the proteome? What is the conceptual difference or advance relative to previous studies? The authors should discuss these differences and mention the previous work earlier in the results part.

Text:

line 31: better "rate" instead of "rhythm", because "rhythm" cannot be constant.

line 32: How can codons per se be scarce?

line 51: Better "because" instead of "although"

line 55: "shielded" or "obscured" instead of "cryptic"

line 69: residues cannot be D/E rich

line 77: The term "noncanonical translation" seems to be too broad to be checked by a single reporter assay

line 113: Why should be an purified translation system richer in "transacting factors" than a model system in vivo? Which transacting factors?

Figures:

Figure 1B: Are SD values estimated based on technical or biological replicates? What are "trials"?

Figure 1C: Description of error bars is missing in the Figure caption.

Figure 1F: What is shown, technical or biological replicates?

Figure 1G: What is shown, technical or biological replicates?

Figure 3E: The authors extracted the monoisotopic masses. Skyline could be used to extract the complete ion pattern (1), (2) smooth the data and make it visually more appealing and (3) provide ion dot products.

Why is this experiment considered to prove the existence of peptidyl-tRNA? If the lysate was analyzed, all abortion products (whether attached to a tRNA or not) would be detected.

Figure 4A: As the mass spectrometric analysis detects only peptides and not peptidyl-tRNAs, the label in the schematics should be changed accordingly.

Figure 4B: The indicated number of wt and Δ pth2 specific peptides is almost equal. Why wt-specific peptides are so highly abundant?

Figure 4E: black arrows are not explained.

Figure S3C: Is it possible to annotate more prominent fragments? Are internal fragments or a-ions observable? In the figure caption, "fragment" is used with two meanings (mass spec and premature termination). Is it possible to avoid this nomenclature ambiguity?

Figure S3D: Peptide sequences in the Figure seem wrong, as they should depict Asn strings (and not Asp). Monoisotopic masses also seem to refer to Asp strings.

+ Typo in the Figure legend "aare shown"

Figure S4: The X-axis seems to be truncated. Why not all 682 peptides are depicted?

Reviewer #1:

Previous work from the same group identified a phenomenon in bacteria whereby translation of polyacidic sequences causes premature peptidyl-tRNA release. This process was termed intrinsic ribosome destabilization or IRD for short. The authors proposed that the polyacidic sequences caused a destabilization of the ribosome that was counteracted to some extent by ribosomal protein L31 as well as by increasing the magnesium concentration.

The manuscript of Ito et al addresses the question as to whether this process also occurs in eukaryotes, which is a reasonable assumption given the conservation of the translational apparatus. Using a series of in vivo and in vitro assays the authors present convincing evidence that this is indeed the case and identify a number of endogenous proteins that undergo IRD in yeast. Overall, I find the manuscript to be very nicely written and presented. As far as my experience allows (i.e. I am not a mass spectrometry expert), the experiments appear to be technically well performed and appropriately interpreted and the results support the conclusions. In my opinion the main weakness of the manuscript (and perhaps also for the previous bacterial version) is related to the mechanism and order of events (see points below). Any additions in this direction would certainly strengthen the manuscript.

We are grateful for the reviewer's positive evaluation and the following constructive comments.

1. I am still a little confused by the order of events in this IRD process, even in relation to the bacterial system. Does the polyacidic peptidyl-tRNA cause subunit splitting (which is the destabilization of the ribosome that is always talked about?) and this is the cause of the peptidyl-tRNA drop-off? If the authors want to use the term IRD for eukaryotes then shouldn't they actually show that subunit splitting by polyacidic peptidyl-tRNA is also occurring in the eukaryotic system?

We agree that the molecular mechanism was vague in the previous manuscript. As detailed in the response to the point 2 below, additional experiments have partially clarified several points regarding mechanisms. Briefly, the polyacidic peptidyl-tRNA was still associated with the 80S ribosome complex in a translationally inactivated state, where Pth could cleave the 80S-associated peptidyl-tRNA (new Fig. 2). We hypothesize that polyacidic peptidyl-tRNA somehow alters the conformation of the ribosome, exposing the peptidyl-tRNA to Pth.

Also, we note that there has been significant progress on the molecular mechanism of IRD in *E. coli*. Even in *E. coli*, the translation of negatively charged amino acids alters the conformation of the translating 70S ribosome, rendering it in an interrupted state. These novel findings in both eukaryotes and *E. coli* imply that translation of polyacidic sequences alters the ribosome into a previously unknown suspended state before premature termination eventually occurs. Together with other similarities in the phenomenological aspects (D/E-enrichment, N-

terminal bias, Mg²⁺ dependency, stochastic feature, translation discontinuation rescued by Pth, and so on), we consider that it is safe to designate this phenomenon in eukaryotes as IRD.

Finally, further mechanistic analysis is of course interesting, but we would like to scope the analysis in our next work.

2. In relation to point 1, are there actually experiments showing that the peptidyl-tRNA is actually dissociating from the ribosome and this is the substrate for the PTH? I don't see any experiments excluding the possibility that the PTH can actually cleave the peptidyl-tRNA while its still bound to the large subunit (but after splitting). The manuscript lacks any summary model schematic figure...the previous bacterial paper had a summary only in the graphical abstract which suggested splitting but was very vague in terms of the order of events.

In response to comments raised by this reviewer and others, we conducted additional experiments using the human PURE system to investigate how polyacidic peptidyl-tRNA actually affects the conformation of the translating ribosome. A series of experiments including sucrose density gradient (SDG) centrifugation, toeprint analysis, and chase experiment have revealed the following about the molecular mechanism.

- i) SDG analysis: we detected the 10D peptidyl-tRNA in the 80S ribosome fraction (new Fig. 2B), indicating that at least a fraction of eukaryotic ribosomes retained the 80S complex during the translation of a polyacidic sequence. In addition, Pth2p treatment diminished the co-migration of the peptidyl-tRNA and the 80S ribosome. These results suggest that the 10D peptidyl-tRNA within the 80S complex is exposed in a Pth2p-accessible state.
- ii) Toeprint analysis: we observed several reverse transcripts, indicating the stalled ribosomes on the 10D mRNA sequence. These reverse transcripts disappeared upon the Pth2p treatment. Taken together with the SDG analysis, we assume that the 10D-run-translating ribosomes maintain the 80S complex.
- iii) Chase experiment: ribosomes translating the D/E runs were retained as the peptidyl-tRNA-accommodated complexes for a long time, under the conditions where the ribosomes paused by a polyproline translation resumed the elongation.

From these results, we speculate that the translation of the polyacidic sequences somehow changes the conformation of the 80S complex into a suspended state, eventually leading to a stochastic abortion.

These results have been included as the new Fig. 2. We hope the reviewers agree with the model suggested by these results.

3. The bias against polyacidic sequences in the N-terminal 30aa appears relatively modest. One wonders why these sequences are not simply selected against if they induce IRD i.e. are

these sequences actually conserved in these proteins across different organisms (eukaryotes, but maybe archaea and bacteria too?). Somehow this reminds me of the polyproline situation except here life evolved a solution e.g. EF-P and eIF5A so as to be able to maintain the presence of polyproline sequences in proteins. But here this is not the case. If I remember correctly, there was only one protein Val-tRNA synthetase that has a conserved PPP motif across all kingdoms of life. How is it for polyacidic sequences?

Thanks for raising this important point from an evolutionary perspective. It would be very interesting if there were some genes with well-conserved acidic residues in the N-terminal regions. However, we could not find any conserved N-terminal D/E-rich sequences among the ORFs that induced the polyacidic residues-dependent premature termination. Anyway, this point is very important to unveil the physiological significances of IRD, so we would like to continue this wider analysis in the future.

Reviewer #2:

This manuscript outlines an effect that has been discovered by the same group about 5 years ago and had not been shown to exist in eukaryotes before. In the light of recent great interest in translation, failure of translation, and translation quality control, I believe that this finding is of high interest to the wider community. The manuscript also introduces a novel methodology to identify peptides from peptidyl-tRNAs via mass spectrometry.

In my opinion, the experiments were performed to a high standard and support the messages well.

I found the manuscript well written, easy to follow, and enjoyed reading it. The methods are generally well described. The figures are well designed, as well.

We are very grateful for this positive evaluation.

1) In Fig. 1 I found it difficult to compare TC indices between panel B and panel C due to the difference in representation (horizontal vs. vertical).

According to the comments, we modified the representation of Figure 1B.

2) Why are the results different between Fig. 1F and 1G for the GFP5-10D without added Mg²⁺? (~0.18 vs. ~0.05) Am I correct that in Fig 1F 0 mM added Mg²⁺ was applied?

We appreciate your pointing this out. We noticed that the formula used in Fig. 1F was incorrect and replaced it with the corrected data. The recalculated values were largely consistent with those in the new Fig. 2A (previous Fig. 1G).

3) Would it make sense to speculate why RTF1 has a much lower TC than ANB1 and NMD2? Could it be because the all 4 D/Es are almost in a single stretch with only 1 other amino acid between?

As the reviewer speculated, it is likely that the high density of the D/Es in RTF1 could contribute to the lower TC than ANB1 and NMD2.

4) Is there any hint, e.g., in the literature that endogenous IRD targets, such as JIP3 are upregulated upon high Mg²⁺ concentrations in the cell? To my knowledge this would occur, e.g., upon ATP depletion.

Thank you for the suggestion. Unfortunately, our survey of literature has so far failed to find any such examples. However, since this is indeed an intriguing possibility leading to the biology of IRD, we will continue to survey this issue.

5) Are D/E rich N-termini of endogenous IRD targets conserved across species?

According to the comment, we conducted an additional bioinformatics analysis to determine whether polyacidic sequences in the N-terminal regions are actually conserved across different organisms. However, we could not find any conserved N-terminal D/E-rich sequences among the ORFs that induced the polyacidic residue-dependent premature termination.

6) Fig. 4 Are non D/E ending peptides generally enriched in D/E above average?

According to the suggestion, we analyzed the MS data updated in the revision and found that the D/E enrichment of the *pth2Δ*-specific C-terminal non-K/R peptides was statistically higher than average. We included this result in the revision (new Fig. 5F, G, Supplementary Fig. S5E, F).

7) Related to 6: Could the D/E rich C-terminus of peptidyl-tRNA peptides have been lost due to LysC cleavage and poor flyability? If you analyze the peptide composition following the K/R cleavage site, is there an enrichment in D/E?

The reviewer's prediction is correct. We found examples of D/E-rich sequences downstream of the *pth2Δ*-specific C-terminal K/R peptides. In addition, the frequency of the D/E-enriched

cluster in the *pth2Δ*-specific K/R peptides was statistically higher than the others. We included this result in the new Supplementary Fig. S5G, H.

8) I found the graph in Figure S6B hard to understand. I believe the curve should not be smoothed as there are 15 discrete steps from 0 to 15 D/E residues, so it should look like a step graph.

The supplementary figure was indeed hard to understand. Since the figure does not add new information and thus is not essential, we omitted the figure in the revision.

9) In Figure S7 the region in which D/E residues are disfavored appears to be a lot shorter than in yeast. I believe this is noteworthy and should be commented. It might also be very interesting to repeat the analysis for a few more different eukaryotes, to check if this is something that is, e.g., common to mammals or similar.

According to this comment and the comments from the other reviewers, we conducted an informatics analysis to determine the D/E usage biases in the proteomes of other organisms in all kingdoms of life. The overall trend, including the human proteome, was to disfavor the D/E-rich sequences in the N-terminal regions, although the degrees and distances of the bias varied. However, there were proteomes with no apparent bias, such as *Arabidopsis thaliana* and the archaeon *Pyrococcus furiosus*. These results are included in the new Supplementary Fig. S7.

10) I was wondering which datasets were inspected in the GWIPS viz analysis. This made me realize that I did not find any description of this analysis in the Methods part. I believe it should be added.

We inspected the Ribo-seq data of Vhs3p and others in the “global aggregates” of GWIPS viz. We added the description in the Results, as shown below.

The continuous translation after the long D/E runs seen in Vhs3p and others was confirmed by the GWIPS-viz browser (as global aggregate) 44, a public web tool to visualize the ribosome distribution.

11) In the description of how the rolling-window analyses of D/E richness was done, it is described that all genes above 130 amino acids length were separated into 100 amino acid N-terminus, 100 amino acid middle and 100 amino acid C-terminus. Should this not only work for genes of at least 300 amino acids length? What happened for those shorter? Were they excluded or were 3 overlapping regions included in the analysis?

As the reviewer pointed out, the N-terminal, middle, and C-terminal datasets include the overlaps derived from ORFs shorter than 300 aa. However, excluding ORFs shorter than 300 aa severely limited the population of the analysis (130 aa: 680 ORFs; 300 aa: 2169 / 6050 ORFs are omitted) and thus could have introduced another bias. Therefore, we used the current definition in this analysis.

12) In l. 280-282 it is written:

“Strikingly, we found that the number of proteins with ≥ 3 D/E residues in the N-terminal regions dropped early, from the N terminus to 30-40 amino acids, which is similar to the length of the tunnel.”

I believe that the term “dropped” is not the best to describe this, as this implies that the number of such proteins is high at first and then goes down. For what I see, this is not the case. A better term might be “is low”, but there might be better ways to describe this. Consider re-phrasing this sentence.

We rephrased the term from “dropped” to “was low” or “low frequency”.

13) l. 390-392:

“Second, the comprehensive identification of the entire peptidyl-tRNAs is hampered by technical limitations of MS, such as low ionization of the peptides and insufficient sensitivity.” I would argue that it is not the entire peptidyl-tRNA but “peptidyl-tRNA derived peptide”, as the tRNA is cleaved off prior to mass spectrometry. Therefore, one might consider rephrasing, possibly also at other points in the paragraph to ensure it is clear that only peptides are analysed, not the tRNA part.

We agree. We rephrased the wording in the corresponding section and the figure.

14) l. 536-537:

“In this analysis, every ten peptides with high reproducibility were selected and used for comparison.” I do not fully understand what is meant by this sentence. Can you explain a bit more?

Since the figure does not add new information and thus is not essential, we omitted the figure in the revision.

15) Figure caption 1E) “Radioactive bands were detected by a phosphorimager.” Technically they were surely detected with the help of storage phosphor screens?

We corrected the description as below.

Radioactive bands were developed by using an imaging plate.

16) It would be good to mention in the abstract that most of the study was done on S. cerevisiae.

It is true that the results on *S. cerevisiae* would be critical throughout the study, but we believe that the study of the human-factor based reconstituted translation system (Hs PURE system) also contributes significantly. Therefore, we mentioned both budding yeast and human in the abstract.

17) Table S1: There is a strain named SCYXXX. I believe this has to be named.

Thank you for pointing this out. We replaced it with the corrected name.

Reviewer #3:

Taguchi and colleagues have previously determined that translation of an mRNA in bacteria is aborted due to intrinsic ribosome destabilization (IRD) if the open reading frame (ORF) contains consecutive codons at or near its 5'-end that encode negatively charged amino acid residues. Here, these researchers report the results of a logical and well-executed series of experiments showing that IRD also occurs in eukaryotes, and that the dependence of IRD on specific sequences near the N-terminus of polypeptides leads to selection against their occurrence in eukaryotic proteomes. The finding that the peptidyl-tRNA hydrolase Pth2p is involved in the cleavage of the abortive polypeptidyl-tRNAs generated by IRD suggest that they are released from destabilized ribosomes: the occurrence of IRD in a fully in vitro reconstituted human translation system that contained only ribosomes, tRNAs, elongation factors, termination factors and aminoacyl-tRNA synthetases excluded the involvement of known recycling factors in the splitting of ribosomes.

However, splitting of arrested ribosomes into subunits was not shown, and the mechanism of IRD remains unknown. Although the authors do not explicitly state how they propose that nascent peptidyl-tRNAs are released from ribosomes following termination, line 387 suggests drop-off. Notably, the authors also showed that elevation of Mg concentrations alleviates IRD and allows uninterrupted translation of proteins containing N-terminal D/E-rich stretches. It would add to the report if the authors tested whether arrested ribosomes could resume translation after elevation of Mg concentrations and if they could, whether such an ability depends on the time of initial incubation at low Mg concentrations that are permissible for IRD. These experiments can be done in the in vitro reconstituted system and resumption of translation can be monitored by disappearance of peptidyl-tRNAs. The proposed experiments could potentially estimate how quickly presumable drop-off of peptidyl-tRNA occurs in the reconstituted system.

We greatly appreciate the positive evaluation and proposed experiments described in the latter part of this comment.

We agree that the proposed experiment using the reconstituted system will provide critical insight into the mechanism of this phenomenon. The chase experiment revealed that the ribosomes translating the D/E runs remained as the peptidyl-tRNA-accommodated complexes for a long time, under the conditions where the ribosomes paused by polyproline translation resumed the elongation. This result was unexpected, but combined with the density gradient centrifugation and toeprint analysis (see above for our response to the comment 2 by reviewer #1), these results suggest that polyacidic peptidyl-tRNA alters the conformation of the ribosome into a suspended state, eventually aborting translation. The results are included in the revision as the new Fig. 2.

1. It would be helpful to readers if the authors explained why they included the CrPV IGR IRES in their mRNAs.

Thank you for the suggestion. We added the description, as shown below.

We constructed fusion genes encoding GFP (GFP₅, GFP₃₀, or GFP_{FL})-10X-Fluc (N-terminal 84 a.a. fragment). CrPV-IRES, which allows translation initiation without initiation factors, was used to minimize the length of the peptide translated prior to the D/E-runs.

2. Typographical and other minor errors.

Line 135. It would be more accurate to state that additional Mg⁺⁺ partially alleviated IRD.

We corrected the wording.

Line 154. The term ‘immaturely’ should be replaced by ‘prematurely’.

We replaced the term accordingly.

Figure 1G. Why “+/-“ 0?

We modified the description as “+0”.

Lines 184-185. Verma et al (Ref. 36) claimed that Vms1p is a peptidyl-tRNA hydrolase and not an endonuclease, so that this description is inaccurate.

According to the studies by Kuroha *et al.* and Sichen *et al.*, Vms1p cleaves the CCA acceptor of peptidyl-tRNA to generate a peptidyl-CCA molecule. We therefore recognize Vms1p as a tRNA endonuclease and have added the corresponding reference and modified the description.

Reviewer #4:

Major points:

Conceptual clarity:

1. While the manuscript shows convincingly that translation is terminated by specific acid amino acid sequences and the respective peptidyl tRNAs accumulates in the cell, the exact mechanism of the abortion is not entirely clear. Peptidyl transfer of highly acidic sequences could be slowed down kinetically favoring the drop-off of the peptidyl-tRNA from the ribosome. Second, highly acidic nascent chains could destabilize the peptidyl-tRNA leading to their dissociation. Third, highly acidic nascent chains could destabilize the ribosome leading to IRD (intrinsic ribosome destabilization) events as proposed recently (Chadani et al., 2017). How do the authors discriminate these scenarios? What is the basis for the conclusion that highly acidic sequences destabilize the ribosome? Increasing Mg²⁺ concentrations (e.g. Fig1G & S1E) of course stabilize the subunit association but also many other RNA interactions such as the binding of the peptidyl-tRNA to the P site. How do the authors discriminate peptidyl-tRNA drop-off from the ribosome from an IRD event? Is there biochemical or biophysical evidence that ribosomes are indeed dissociated? How can the length effect of the nascent peptidyl chain can be interpreted in the light of IRD events? While it is easy to understand that long peptidyl-tRNA do not drop off from the ribosome, it is unclear why they should not dissociate the ribosome in IRD events (as seen in the E. coli system). As the motivation for the paper was to understand “whether IRD takes place even in eukaryotic cells” (line 57), in particular because the eukaryotic ribosome association appears to be more robust than the prokaryotic one (line 351), these controls have to be provided. Referring to the high conservation of ribosome dynamics among different organisms (line 316) and the high conservation of the ribosome exit tunnel (line 318) is not sufficient. The authors should sharpen their mechanistic conclusions and provide essential controls that show that these sequences indeed destabilize the ribosome itself.

We agree that the mechanistic aspect was lacking in the previous version. Therefore, we conducted several additional experiments to shed some light on the mechanism: sucrose density gradient centrifugation, toeprint analysis, and chase experiments. These analyses contributed to the clarification of some points of the molecular mechanism. These results indicated that a highly acidic peptidyl-tRNA somehow alters the conformation of the 80S ribosome into a suspended state, eventually aborting the translation. Importantly, Pth2p can access the peptidyl-tRNAs that are associated with the 80S ribosome, implying that the polyacidic peptidyl-tRNA impairs the integrity of the ribosome to expose the peptidyl-tRNA. We consider that it is safe to designate this phenomenon in eukaryotes as IRD.

2. The authors state that they find “many endogenous D/E-rich peptidyl-tRNAs... in cells lacking a peptidyl-tRNA hydrolase”. However, in the corresponding Figure 4 and Figure S4

peptides are categorized based on their C-terminal amino acid. Using this definition AVNELPTKSTD (Figure 4E) is D/E rich while ADESDPVESK (Figure S4B) is not D/E rich. What is the rationale for this definition? If the author would like to conclude that the PTH-dependent peptides are D/E rich, they should analyze the D/E enrichment as well.

In the previous Fig. 4E and Fig. S4B, only peptides with D/E at the C-termini were highlighted in red, where the highlighted peptides were not necessarily rich in D/E, as the reviewer suspected. Since this was not suitable to show the D/E-rich property, we have changed the presentation to express the D/E enrichment by showing it in a color gradation (new Fig. 5G and Supplementary Fig. S5F). Also, we included boxplots to quantitatively show the D/E-enrichment in the peptidyl-tRNA-derived peptides (new Fig. 5F and Supplementary Fig. 5E).

Data presentation & validation: Figure 4

3. The MS data are difficult to evaluate. Deposited files are named Ito1, Ito2, etc. The submitted peptides and protein groups files do not contain the data depicted in Figure 4. The parameters of the MaxQuant search are not comprehensively described in the Method section (FDR, variable modifications...). The authors should provide a more user-friendly, better annotated data access.

We entirely analyzed the MS data again with the Proteome Discoverer 2.4 software instead of MaxQuant. The basic settings of the software are described in the Methods section and the comprehensive settings are listed in Supplementary Table S4. In addition, we attached the list of the WT-specific and the *pth2* Δ -specific peptides as Supplementary Dataset 1. We have also appended raw data files in the repository (jPOST).

Preview code: <https://repository.jpostdb.org/preview/976174466303300963cfd> (Access key: 4264)

*4. The enrichment of nascent chains by phenolisation and their subsequent mass spectrometric detection is elegant and relies on well-established analysis workflows. The enrichment of acidic sequences in Δ *pth2* cells and their N-terminal location is convincing and in agreement with the translation experiments in vivo and in vitro. However, the analysis is qualitative only. It is unclear whether the experiment was replicated, as only data for one biological replicate was deposited. The nature of the technical replicates is unclear as well. The authors should state how the experiment was replicated. Given that the authors present the assay for the first time, replicates would help to judge the robustness of the results and the enrichment of peptidyl-tRNAs.*

As mentioned in a prior response to the reviewer's other comments, we reacquired all MS data and reanalyzed them for this revision. We now show two biological replicates in the new Fig. 5. In addition, we conducted the MS analysis using the lysates from other growth phases (new

Supplementary Fig. S5). All the MS data have clearly shown the D/E-enriched peptides derived from the peptidyl-tRNAs in the *pth2Δ* lysate. Taken together, we believe the new MS data are very robust and further reinforce our claims.

5. The authors state that they aimed to “broadly explore the ORFs that prematurely terminate the translation”. The analysis revealed 36 target peptides originating from an even lower number of proteins. The authors conclude that the occurrence of drop-off products might be underestimated and that technical limitations such as low ionization of target peptides and insufficient sensitivity hamper the analysis (line 391). It is unlikely that in the analysis of yeast lysates the sensitivity of MS itself is limiting, because the amount of available sample should not be limiting and the authors could have loaded more peptide digest per analysis. Inspecting the RAW files indeed shows that the authors were able to saturate the LC-MS system. Rather, a combination of the only moderate enrichment of target peptides (all less than 100 intense compared with the most prominent peptides), the high concentration of lab contaminants and the limited dynamic range of MS presumably led to the limited number of hits. As the authors state, the discovery of the target peptides is likely impaired by their poor ionization. However, if a more comprehensive analysis of abortion products would be desired, biological replicates, prefractionation on the peptidyl-tRNA level (e.g. anion exchange separation) or on the peptide level (reverse phase or size exclusion), in-gas fractionation or simply longer LC runs (currently only 30 min gradients) could likely boost the depth of analysis. Because trypsin/LysC treatment leads to poorly ionizing semi tryptic peptides, LysN/ArgN or TrypsinN might be better options. They should produce better ionizing peptides and lead to cleaner MS/MS spectra due to more complete b-ion ladders. Both should result in a higher observability and better peptide scores. If the authors are satisfied with the current depth of analysis, they should better express the qualitative nature of the experiment. Alternatively, the authors might apply one of the commonly used strategies to overcome these technical issues. This would allow to better estimate how frequently these abortion events occur.

According to the reviewer’s suggestion, we tried increasing the sample volume from a 200 mL culture to 2 L culture. Although the signal intensity increased, we detected many contaminant proteins/peptides and could not improve the detection of *pth2Δ*-specific peptides. This is probably because it is difficult to maintain the efficiency in the separation by acid phenol. In other words, the problem is not only the MS preparation/measurements but also the basic biochemical experiments, especially the acid phenol extraction.

The reviewer kindly provided various suggestions for improving the MS detection; however, due to the difficulties in principle (separation of pep-tRNAs by anion exchange) and limitations of the equipment (fractionation at the peptide level by HPLC and in-gas fractionation before ionization), we have only adopted a longer gradient time (from 30 min to 70 min). We also tried using Lys-N or Tryp-N, but we could not find an appropriate digestion

conditions in our lab.

6. I assume that tRNAs connected to longer peptidyl-chains or more hydrophobic sequences are more likely located in the interphase after phenol extraction. How do the authors consider such potential bias towards the isolation of short, hydrophilic peptidyl-RNAs?

As the reviewer pointed out, the length and hydrophilic/hydrophobic properties of peptidyl-tRNAs cause a bias for the partitioning of peptidyl-tRNAs into water and interphase. On the other hand, it is true that the analysis of the interphase would be very interesting, but it is hard to identify *pth2Δ*-specific peptidyl-tRNAs among large amounts of endogenous elongating peptidyl-tRNAs and full-length proteins. On the other hand, although the aqueous phase has a bias toward peptidyl-tRNAs with relatively short peptide moieties, it is more favorable for abortive peptidyl-tRNAs, which are the target of our analysis.

Minor points:

1. The authors should discuss their results in the light of the available literature on the effect of highly acidic sequences in the PTC (e.g. Leininger et al., 2021; PMID: 34652913).

We add the description of the literature in the Discussion section.

2. Given that the cellular localization of Pth2 seems to be under debate, the authors should discuss the cellular localization of peptidyl-tRNAs that are enriched.

We added the sentences for the localization of the abortive peptidyl-tRNAs as follows.

The observations that *E. coli* Pth rescues the lethal phenotype of *pth2Δ* upon the expression of D/E-runs and the association of the polyacidic peptidyl-tRNA with the 80S ribosome complex support our assumption. However, more careful analyses on the localization of Pth2p should be done in the future.

3. The bioinformatics analysis revealed that negatively charged amino acids are avoided in the N-terminal part of proteins. At the end of the discussion the authors mention that this is consistent with previous reports (line 403). Why was it necessary to reanalyze the proteome? What is the conceptual difference or advance relative to previous studies? The authors should discuss these differences and mention the previous work earlier in the results part.

In the previous manuscript, it is true that those earlier bioinformatics analyses were only mentioned at the end of the Discussion, so we agree that it would be fair to introduce them earlier, in the Results. In terms of content, the analyses by other groups are not exactly the same because of different definitions and the analyses of only the human and budding yeast proteomes. Our previous analysis only showed the bias found in bacteria. In this respect, it

would be worth showing the data about many organisms analyzed using the same definition in this study.

Text:

line 31: better “rate” instead of “rhythm”, because “rhythm” cannot be constant.

We corrected the wording.

line 32: How can codons per se be scarce?

It is indeed true. We have corrected “the scarcity of codons or tRNAs” to “the scarcity of tRNAs in corresponding codons”.

line 51: Better “because” instead of “although”

We corrected the wording.

line 55: “shielded” or “obscured” instead of “cryptic”

We corrected the wording.

line 69: residues cannot be D/E rich

We have corrected “D/E-rich residues” to “D/E-rich sequences”.

line 77: The term “noncanonical translation” seems to be too broad to be checked by a single reporter assay

We have corrected “noncanonical translation” to “such premature translation termination”.

line 113: Why should be an purified translation system richer in “transacting factors” than a model system in vivo? Which transacting factors?

Our previous description was ambiguous. In the revision, we have amended this part as follows.

In *E. coli*, a reconstituted cell-free translation system (*E. coli* PURE system) was used to demonstrate that the polyacidic residue-dependent premature termination occurs only with the minimum factors required for translation.

Figures:

Figure 1B: Are SD values estimated based on technical or biological replicates? What are “trials”?

SD values were estimated as biological replicates for the *in vivo* reporter assay and technical replicates for the *in vitro* translation assay. We added descriptions in the corresponding sections.

Figure 1C: Description of error bars is missing in the Figure caption.

Figure 1F: What is shown, technical or biological replicates?

Figure 1G: What is shown, technical or biological replicates?

We added the description in the corresponding sections.

Figure 3E: The authors extracted the monoisotopic masses. Skyline could be used to extract the complete ion pattern (1), (2) smooth the data and make it visually more appealing and (3) provide ion dot products.

We reanalyzed these data with the Skyline software and replaced them with new graphs in the revised manuscript. Isotope dot product (idotp) scores are also shown in the revised figure.

Why is this experiment considered to prove the existence of peptidyl-tRNA? If the lysate was analyzed, all abortion products (whether attached to a tRNA or not) would be detected.

Our description was inaccurate. The fraction we analyzed in this MS analysis (Fig. 4E) was not simply the lysate, but a total RNA fraction after the phenol extraction as in Northern blotting (Fig. 4C, D). We modified the corresponding sentence as follows.

The MS analysis of the phenol-extracted RNA fraction of the *pth2Δ* strain expressing GFP_{5-10D} identified several unique peptides derived from the peptidyl-tRNAs.

Figure 4A: As the mass spectrometric analysis detects only peptides and not peptidyl-tRNAs, the label in the schematics should be changed accordingly.

We modified the figure (new Fig. 5A) accordingly.

*Figure 4B: The indicated number of wt and Δ*pth2* specific peptides is almost equal. Why wt-specific peptides are so highly abundant?*

We repeated this analysis with another culture condition (mid-log culture). From these results, the number of WT-specific peptides was decreased (Fig. 5B and Fig. S5A). The re-analysis from the late-log culture lysate (replicate 2) also showed similar tendencies, so we suspect that many WT-specific peptides would be derived from contaminant proteins that were present during the phenol separation. As we discussed above, the acid-phenol extraction does not completely separate peptidyl-tRNAs from proteins/peptides, so it is difficult to eliminate such contaminating peptides completely.

Figure 4E: black arrows are not explained.

In the revised manuscript, we deleted the black arrows in the corresponding figure (new Fig. 5G; previous Fig. 4E).

Figure S3C: Is it possible to annotate more prominent fragments? Are internal fragments or a-ions observable? In the figure caption, “fragment” is used with two meanings (mass spec and premature termination). Is it possible to avoid this nomenclature ambiguity?

Since we used HCD/CID for generating the fragments for MS/MS, we consider only b- and y-ions should be used for the annotation. Note that these annotations were obtained from the automatic search by the software. The term “fragment” with the two meanings was confusing. Then, we have not used this term for the peptide “fragment” in the corresponding legend (new Fig. S4C) in the revised manuscript.

The MS/MS spectrum of the M_{ox}VSIGDDDDDDDD peptide expressed from GFP_{5-10D}. The peaks of the b- and y- ion fragments are shown in red and blue, respectively.

Figure S3D: Peptide sequences in the Figure seem wrong, as they should depict Asn strings (and not Asp). Monoisotopic masses also seem to refer to Asp strings.

We corrected this mistake.

+ Typo in the Figure legend “aare shown”

We corrected the typo.

Figure S4: The X-axis seems to be truncated. Why not all 682 peptides are depicted?

We decided not to use this figure presentation in the revised version. For reference, the reason why we chose the X-axis up to 500 aa in this Supplemental figure is because the X-axis of the corresponding main figure was up to 500 aa.

REVIEWER COMMENTS

Reviewer #1 (Remarks to the Author):

The authors have addressed my comments satisfactorily. The important point was to provide some more mechanistic insight into the IRD process in eukaryotes. To do this the authors have run reactions on a sucrose gradient showing that the GFP-10D-stalled ribosomes remain as 80S i.e. the 10D does not lead to splitting of the 80S subunits and that Pth2p appears to access the 80S to hydrolyze the peptidyl-tRNA on the 80S ribosomes. These are exciting findings that strengthen the paper considerably.

Reviewer #2 (Remarks to the Author):

Comments for "Nascent peptide-induced translation discontinuation in eukaryotes impacts biased amino acid usage in proteomes"

Essentially, all my concerns have been addressed (at least in the response letter), although not all additional analyses and explanations have made their way into the current version of the manuscript. However, I consider this as fair, judging by the amount of additional information added to the manuscript now.

From my side, there are no further concerns about questions I raised before and I am looking forward to seeing the work published.

Reviewer #4 (Remarks to the Author):

In general, I appreciate the efforts of the authors to get additional insights into the mechanism, to broaden the MS analysis and to improve the MS data presentation. The new experiments are insightful and the story is interesting for a broad audience. However, the conceptual clarity and mass spectrometry issues have only partially been addressed and improved. Thus, prior to publication the remaining issues have to be resolved:

Regarding the major points raised previously:

Point1: Conceptual clarity

Notably, the story of the paper has changed significantly compared to the first manuscript. If I understand the new storyline correctly, Poly-E/D sequences stop translation and inactivate ribosomes. While this leads to ribosome destabilization and dissociation in prokaryotes, eukaryotic ribosomes are stalled and recycled in a subsequent QC event by the peptidyl-tRNA hydrolase, which is recruited to the ribosome.

Previously, the authors discovered a process that dissociates ribosomes in prokaryotes and termed it "intrinsic ribosome destabilization". In eukaryotes, similar sequences lead to ribosome stalling but not ribosome dissociation. Why should stalled complexes that are stable over hours, be termed destabilized? For comparison, Poly-Pro sequences stall and strain parts of the ribosomes (Pro-stalled complexes are even less stable than Poly-Glu complexes according to the current manuscript). Features of Poly-Pro stalled complexes then recruit EF-P. However, Poly-Pro stalling is generally not considered as destabilizing event. Also, stop codons that recruit termination factors and lead to termination do not destabilize the ribosome. Unfortunately, my previous questions on complex stability have not been addressed (interpretation of the Mg-dependence, the discrimination between tRNA drop off and ribosome destabilization and the interpretation of the nascent chain length effect). Why the magnesium effect cannot be explained by a reduced fidelity of translation or a stabilization of the peptidyl-tRNA? If the authors would like to call the process "IRD event", emphasizing ribosome destabilization, the questions should be addressed and ribosome destabilization should be shown. In eukaryotes, the D/E enrichment, N-terminal bias, Mg dependence, stochastic feature and rescue by

Pth show strong similarities to the prokaryotic system (presumably because the underlying stalling events are similar), but they do not show a ribosome destabilization. If "Intrinsic ribosome destabilization" does not mean destabilization of the ribosome (as in prokaryotes), the authors should explain their terminology better (also in the abstract) or derive a more intuitive terminology.

In different parts of the text, the authors state that IRD-derived abortive peptidyl-tRNAs accumulate (e.g. lines 198,228, 251, 253 and depicted in Figure 5a). In the light of the new experiments and the model, it is unclear to me whether and to which extent peptidyl-tRNAs accumulate off the ribosome. Perhaps the authors could explain how ribosome-bound and free peptidyl-tRNA were discriminated.

In lines 149-170 the authors describe that the Glu-stalled complexes are recognized by PTH and subsequently hydrolyzed, while in line 349 the authors conclude that "only essential factors are sufficient for premature termination". How both can be the case?

In line 344 the authors write that the integrity of the ribosome seems impaired because abortive peptidyl-tRNAs are produced, while in line 382 the authors conclude that eukaryotic ribosomes can maintain the 80S complex.

Overall, the authors should sharpen their mechanistic conclusions and provide a consistent interpretation throughout the manuscript and in the models (Figure 2F and 5A). If the mechanism is not clear yet, the authors should limit themselves to a more phenotypic description but discuss all mechanistic possibilities.

Point 2: Mass spectrometry

I appreciate the attempts to make the MS search parameters and the data structure more transparent. However, the deposited data overview is unclear, as it does not reflect the real data structure:

- In contradiction to the table, Figure 4B,C,D does not contain MS data.
- In contradiction to the table, Figure 4F,G does not exist
- In contradiction to the table, Figure S4 E,F do not exist, etc...

Moreover, according to the manuscript Skyline and ProteomeDiscoverer were used for MS analysis, while in the reporting summary MaxQuant is the only MS related software.

The authors should clarify these issues to allow for an intuitive reanalysis of the data.

Considering FigS4:

- In contradiction to the reply to the referees, corrections were actually not introduced in the Supplemental Data (FigS4C; formerly S3C).
- Instead of "ion fragments" better use "fragment ions"
- I previously asked for a better annotation of the MS/MS spectrum (FigS4C) and for the extraction of the complete ion pattern together with the associated dot products in Fig4E, because the MS/MS spectrum as it is now (FigS4C) provides only moderate evidence for the identification: 1. The fragmentation pattern annotation in the picture is wrong. The b2 ion comprises three amino acids, while y1 comprises no amino acids. 2. Given that only a minority of fragments is annotated; while many intense fragments remain non-annotated, and given that b4-b8 ions are in the noise and cannot really contribute to the identification, the identification should be substantiated by a more comprehensive annotation. In contrast to the response of the authors, the use of HCD/CID fragmentation does not restrict the nature of ions to y and b ions. Different search engines (e.g. MaxQuant used by the authors) support an expert annotation mode or the annotation could be done manually. The spectrum has at least to contain an a2 ion and fragments related to neutral losses of the methionine. Due to the Asp-stretches (in my experience), internal fragments are also likely. I am happy that the added MS1 filtering (Fig4E) looks nice and that the new ion dot products are supportive. However, I still believe a better annotation of the MS/MS spectrum (and providing the mass deviation of the precursor mass) would help a sound presentation of the data. At least the

fragmentation pattern annotation has to be corrected.

Minor points:

- Line 115: better "even" than "only"
- Wild type abbreviation is sometimes WT and sometimes wt
- *m/z* should be italics
- The different terms "DE usage%" or "DE ratio" are confusing. Perhaps the authors could use one more descriptive term (such as "relative D/E sequence content")
- Perhaps the authors could add a code availability statement

Reviewer #4 :

In general, I appreciate the efforts of the authors to get additional insights into the mechanism, to broaden the MS analysis and to improve the MS data presentation. The new experiments are insightful and the story is interesting for a broad audience. However, the conceptual clarity and mass spectrometry issues have only partially been addressed and improved. Thus, prior to publication the remaining issues have to be resolved:

Regarding the major points raised previously:

We would like to thank the reviewer for further assessment, for appreciating our efforts in the previous revision, and for saying that the story is interesting for a broad audience. We agree that more thorough analyses on conceptual clarity and the MS analysis are merited. Therefore, we conducted additional experiments and further MS analysis, and have amended the manuscript to incorporate these. Please see below.

Point1: Conceptual clarity

Notably, the story of the paper has changed significantly compared to the first manuscript. If I understand the new storyline correctly, Poly-E/D sequences stop translation and inactivate ribosomes. While this leads to ribosome destabilization and dissociation in prokaryotes, eukaryotic ribosomes are stalled and recycled in a subsequent QC event by the peptidyl-tRNA hydrolase, which is recruited to the ribosome.

Previously, the authors discovered a process that dissociates ribosomes in prokaryotes and termed it “intrinsic ribosome destabilization”. In eukaryotes, similar sequences lead to ribosome stalling but not ribosome dissociation. Why should stalled complexes that are stable over hours, be termed destabilized? For comparison, Poly-Pro sequences stall and strain parts of the ribosomes (Pro-stalled complexes are even less stable than Poly-Glu complexes according to the current manuscript). Features of Poly-Pro stalled complexes then recruit EF-P. However, Poly-Pro stalling is generally not considered as destabilizing event. Also, stop codons that recruit termination factors and lead to termination do not destabilize the ribosome. Unfortunately, my previous questions on complex stability have not been addressed (interpretation of the Mg-dependence, the discrimination between tRNA drop off and ribosome destabilization and the interpretation of the nascent chain length effect). Why the magnesium effect cannot be explained by a reduced fidelity of translation or a stabilization of the peptidyl-tRNA? If the authors would like to call the process “IRD event”, emphasizing ribosome destabilization, the questions should be addressed and ribosome destabilization should be shown. In eukaryotes, the D/E enrichment, N-terminal bias, Mg dependence, stochastic

feature and rescue by Pth show strong similarities to the prokaryotic system (presumably because the underlying stalling events are similar), but they do not show a ribosome destabilization. If “Intrinsic ribosome destabilization” does not mean destabilization of the ribosome (as in prokaryotes), the authors should explain their terminology better (also in the abstract) or derive a more intuitive terminology.

Thank you for your deeply insightful comments. We agree to the concern and performed further additional experiments to know whether the polyacidic sequence actually destabilizes the translating ribosome. In this experiment, we fractionated the 80S ribosome translating the GFP₅-10D or GFP₅-10W sequence by SDG in the presence of various concentrations of Mg²⁺ ion. As a result, we found that the 80S accommodating the 10D is more unstable than that translating the 10W at a low Mg²⁺ condition (1 mM), indicating that the translation of polyacidic sequence destabilizes the ribosome complex. The results are included in new Fig. 2C. We assume that the destabilized ribosomes are inactivated, and suspended on the mRNA, but are not necessarily split to the subunit under the reconstituted conditions. We hope the reviewer agrees with us that it may make sense to use “IRD” in the current manuscript.

In different parts of the text, the authors state that IRD-derived abortive peptidyl-tRNAs accumulate (e.g. lines 198,228, 251, 253 and depicted in Figure 5a). In the light of the new experiments and the model, it is unclear to me whether and to which extent peptidyl-tRNAs accumulate off the ribosome. Perhaps the authors could explain how ribosome-bound and free peptidyl-tRNA were discriminated.

We agree that our previous new experiments have raised new questions. In the present data, it is hard to distinguish whether peptidyl-tRNA is inside or outside the ribosome without additional detailed analyses. To address the questions, we need to conduct careful biochemical experiments, including how Pth cleaves the peptidyl-tRNA, and structural analysis using cryo-EM, which, we believe, are the issues we would like to tackle in the near future. In the revised version, we added the statement that the localization of peptidyl-tRNA is currently not clear.

In lines 149-170 the authors describe that the Glu-stalled complexes are recognized by PTH and subsequently hydrolyzed, while in line 349 the authors conclude that “only essential factors are sufficient for premature termination”. How both can be the case?

We interpreted that the referee's concern would be derived from the definition of the word "termination". In canonical translation termination, release factors hydrolyze the peptidyl-tRNA to prevent further elongation. In the case of IRD, the inactivation of the ribosome prior to the hydrolysis of peptidyl-tRNA by Pth2 may be considered as the termination of elongation or, in other words, discontinuation. We agree that the use of the term "termination" can be confusing in some cases. In the revision, we used "discontinuation" in place of "termination" when mechanistic confusion could occur.

In line 344 the authors write that the integrity of the ribosome seems impaired because abortive peptidyl-tRNAs are produced, while in line 382 the authors conclude that eukaryotic ribosomes can maintain the 80S complex.

This point also causes confusion. We added the description in the Discussion section as below.

“However, such inactivated 80S complex is unstable under the Mg^{2+} -depleted condition, indicating that the translation of the D/E runs destabilizes and impairs the eukaryotic ribosome, but does not necessarily split the complex under the *in vitro* conditions.”

Overall, the authors should sharpen their mechanistic conclusions and provide a consistent interpretation throughout the manuscript and in the models (Figure 2F and 5A). If the mechanism is not clear yet, the authors should limit themselves to a more phenotypic description but discuss all mechanistic possibilities.

We thank the reviewer for the overall suggestions. We agree that we should be consistent in our presentation regarding the mechanistic aspects, including the unknowns, throughout the manuscript. In the revised version, we have carefully rephrased the wording. In addition, we have added a schematic of the peptidyl-tRNAs accommodated within the ribosome in Fig. 5A.

Point 2: Mass spectrometry

I appreciate the attempts to make the MS search parameters and the data structure more transparent. However, the deposited data overview is unclear, as it does not reflect the real data structure:

- In contradiction to the table, Figure 4B,C,D does not contain MS data.*
- In contradiction to the table, Figure 4F,G does not exist*
- In contradiction to the table, Figure S4 E,F do not exist, etc...*

We apologize for the confusion. The contents of the repository were not updated for the previous revision. We corrected them appropriately. In addition, to further clarify the data structure, we added the information about the processed data (.pdResult) and the sheet names in the Supplementary Dataset in the corresponding table uploaded in the repository.

Moreover, according to the manuscript Skyline and ProteomeDiscoverer were used for MS analysis, while in the reporting summary MaxQuant is the only MS related software. The authors should clarify these issues to allow for an intuitive reanalysis of the data.

The description in the reporting summary was incorrect. We updated the description about the software.

Considering FigS4:

- In contradiction to the reply to the referees, corrections were actually not introduced in the Supplemental Data (Fig S4C; formerly S3C).*

We apologize for the point. In the revision, we carefully checked that the corrections were introduced in the Supplemental Data.

- Instead of “ion fragments” better use “fragment ions”*

We corrected it.

- I previously asked for a better annotation of the MS/MS spectrum (FigS4C) and for the extraction of the complete ion pattern together with the associated dot products in Fig4E, because the MS/MS spectrum as it is now (FigS4C) provides only moderate evidence for the identification: 1. The fragmentation pattern annotation in the picture is wrong. The b2 ion comprises three amino acids, while y1 comprises no amino acids. 2. Given that only a minority of fragments is annotated; while many intense fragments remain non-annotated, and given that b4-b8 ions are in the noise and cannot really contribute to the identification, the identification should be substantiated by a more comprehensive annotation. In contrast to the response of the authors, the use of HCD/CID fragmentation does not restrict the nature of ions to y and b ions. Different search engines (e.g. MaxQuant used by the authors) support an expert annotation mode or the annotation could be done manually. The spectrum has at least to contain an a2 ion and fragments related to neutral losses of the methionine. Due to the Asp-stretches (in my experience),*

internal fragments are also likely. I am happy that the added MSI filtering (Fig4E) looks nice and that the new ion dot products are supportive. However, I still believe a better annotation of the MS/MS spectrum (and providing the mass deviation of the precursor mass) would help a sound presentation of the data. At least the fragmentation pattern annotation has to be corrected.

We greatly appreciate the reviewers' continued helpful and detailed comments. We did not fully understand the meaning of the previous comment for Fig. S4C (previous Fig. S3C). We re-analyzed the corresponding MS/MS spectrum with Proteome Discoverer 2.4 again and found that many strong signals are derived from the internal fragments of the Asp-stretch, as the reviewer expected. In addition, the strong signal corresponding a_2^+ ion is also annotated. With respect to the neutral losses of oxidized methionine (-CH₃SOH; -63.9983 Da), we annotated manually (defined as < 10 ppm from the corresponding theoretical molecular masses) and found some signals, although their intensities were not strong. The schematic illustration of the detected fragments and figure legend are also corrected appropriately. We believe that the evidence of Fig. S4C is much stronger, thanks to the reviewer's suggestion.

Minor points:

- *Line 115: better “even” than “only”*
- *Wild type abbreviation is sometimes WT and sometimes wt*
- *m/z should be italics*
- *The different terms “DE usage%” or “DE ratio” are confusing. Perhaps the authors could use one more descriptive term (such as “relative D/E sequence content”)*

We have corrected all the points raised by the reviewer.

- *Perhaps the authors could add a code availability statement*

We added a description about the custom R code for the data analysis in the Data availability section.

REVIEWERS' COMMENTS

Reviewer #4 (Remarks to the Author):

I would like to thank the authors for the responsive reply. All my concerns have been addressed. Very nice story.

Minor comments:

The MS/MS spectrum in Fig S4 is more convincing now. However, the fragments DD+ and SI+ are both annotated twice with different masses each.

Moreover, b1+ is annotated. To the best of my knowledge b1 ions are not formed in CID/HCD fragmentation (Ref: PMID: 11180628).

Overall, personally I would remove annotations of fragments that are close to the noise level and thus cannot really contribute to the identification (y3,y4, b1). This might help to tidy up the spectrum and allow to move all annotations in the spectrum. Some fragments are coloured but not annotated.

Reviewer #4 :

I would like to thank the authors for the responsive reply. All my concerns have been addressed. Very nice story.

We thank the reviewer for continuous constructive comments.

Minor comments:

The MS/MS spectrum in Fig S4 is more convincing now. However, the fragments DD+ and SI+ are both annotated twice with different masses each.

Moreover, b1+ is annotated. To the best of my knowledge b1 ions are not formed in CID/HCD fragmentation (Ref: PMID: 11180628).

Overall, personally I would remove annotations of fragments that are close to the noise level and thus cannot really contribute to the identification (y3,y4, b1). This might help to tidy up the spectrum and allow to move all annotations in the spectrum. Some fragments are coloured but not annotated.

According to the suggestion, we have reconsidered the annotations of fragments and then removed b1 and duplicated annotations in the revised figure (Fig. S4C). Regarding the y3 and y4 fragments, y3 and y4 are comparable to the other subtle peaks such as b7 and b8 in our view, so we decided to keep them. We hope this is satisfactory for the reviewer.